# 3,5-Dimethyl-2,4,6-trimethoxychalcone Lessens Obesity and MAFLD in Leptin-Deficient *ob/ob* Mice

**DOI:** 10.3390/ijms25189838

**Published:** 2024-09-11

**Authors:** Stéphanie Gaigé, Anne Abysique, Rym Barbouche, Alain Tonetto, Attilio Di Maio, Maxime Robin, Anh-Tuan Lormier, Jean-Denis Troadec

**Affiliations:** 1Centre de Recherche en Psychologie et Neurosciences (CRPN), UMR CNRS UMR 7077, Aix-Marseille University, 13331 Marseille, France; stephanie.rami@univ-amu.fr (S.G.); anne.abysique@univ-amu.fr (A.A.); rym.barbouche@univ-amu.fr (R.B.); 2PRATIM, FSCM (FR1739), Centrale Marseille, CNRS, Aix-Marseille University, 13397 Marseille, France; alain.tonetto@univ-amu.fr; 3Mediterranean Institute of Marine and Terrestrial Biodiversity and Ecology, IRD, CNRS UMR7263, Aix-Marseille University, 13013 Marseille, France; attilio.dimaio@imbe.fr (A.D.M.); maxime.robin@univ-amu.fr (M.R.); 4Mediterranean Institute of Marine and Terrestrial Biodiversity and Ecology, IRD, NRS UMR7263, Avignon University, 84029 Avignon, France; 5CAYLAB, Contract Research Organization, 13180 Istres, France

**Keywords:** chalcone, food intake, brainstem, hypothalamus, leptin, steatosis, liver

## Abstract

Chalcones constitute an important group of natural compounds abundant in fruits and comestible plants. They are a subject of increasing interest because of their biological activities, including anti-diabetic and anti-obesity effects. The simple chalcone structural scaffold can be modified at multiple sites with different chemical moieties. Here, we generated an artificial chalcone, i.e., 3,5-dimethyl-2,4,6-trimethoxychalcone (TriMetChalc), derived from 2′,4′-Dihydroxy-6′-methoxy-3′,5′-dimethylchalcone (DMC). DMC is a major compound of *Cleistocalyx operculatus*, a plant widely used in Asia for its anti-hyperglycemic activity. Using *ob/ob* mice as an obesity model, we report that, after 3 weeks of *per os* administration, TriMetChalc modified food intake through the specific activation of brain structures dedicated to the regulation of energy balance. TriMetChalc also decreased weight gain, glucose intolerance, and hepatic steatosis. Moreover, through extensive liver lipidomic analysis, we identified TriMetChalc-induced modifications that could contribute to improving the liver status of the animals. Hence, TriMetChalc is a chalcone derivative capable of reducing food intake and the addition of glucose intolerance and hepatic steatosis in a mouse model of obesity. In light of these results, we believe that TriMetChalc action deserves to be more deeply evaluated over longer treatment periods and/or in combination with other chalcones with protective effects on the liver.

## 1. Introduction

Keeping a healthy weight has become both a personal challenge and a major public health issue. The majority of people on the planet actually reside in countries where being overweight or obese is associated with a higher death rate than being underweight. The rate of obesity worldwide has almost tripled since 1975. In 2016, there were approximately 1.9 billion obese adults (those above the age of 18). Of these, more than 650 million were overweight. The World Health Organization has estimated that 39 million children under the age of five and over 340 million children and adolescents will be overweight or obese after 2020 [1]. Fat accumulation in the liver, a defining feature of fatty liver disease, is frequently linked to metabolic syndromes, including obesity. Non-alcoholic fatty liver disease (NAFLD) is a term used to cover a continuum of liver disorders that are characterized by evidence of excessive fat in the liver (hepatic steatosis) and the absence of secondary origins (alcohol consumption, medications, and hereditary disorders). NAFLD is the most common cause of chronic liver disease worldwide and represents a major expanding public health problem [2]. Over one billion people worldwide suffer from NAFLD, which has become the most common form of chronic liver disease. While the majority of people with fatty liver have no or few symptoms, a small subset can develop liver cancer or end-stage liver disease [3]. To better reflect the metabolic-related etiology of this pathology, a new nomenclature was proposed [4,5]. The term metabolic-dysfunction-associated fatty liver disease (MAFLD) is now largely accepted and used. The global epidemic of fatty liver disease is growing alongside the obesity pandemic. The prevalence rate of MAFLD rises with increasing body mass index (BMI). Based on data from 1989 to 2015, the global prevalence of MAFLD in the general population has been estimated to be 25% [6]. The prevalence of MAFLD was 17% in lean individuals and between 50% and 90% in the obese population [7]. For instance, based on histologic analysis of liver originating from automobile crash victim donors, autopsy findings, and clinical liver biopsies, the observed prevalence rates of steatosis are approximately 15% in non-obese persons, 65% in persons with class I and II obesity (BMI 30.0–39.9 kg/m^2^), and 85% in extremely obese patients (BMI ≥ 40 kg/m^2^) [8].

The need for new compounds and possible treatments to prevent the development of obesity and MAFLD persists, despite the wealth of information on the etiology and mechanisms underlying these pathologies. A class of natural flavonoid derivatives of aromatic ketones known as 1,3-diphenyl-2-propen-1-ones, or chalcones, have attracted attention among natural promising chemicals. This interest is based on their numerous biological actions, including anti-obesity and anti-diabetic qualities in both in in vitro and in vivo systems. Many plant-derived forms of chalcones with multiple substitution patterns on both aromatic rings, such as hydroxy, methyl, and methoxy substituents, have been identified [9].

2′,4′-Dihydroxy-6′-methoxy-3′,5′-dimethylchalcone (DMC) is a major compound of *Cleistocalyx operculatus*, a plant that is widely distributed in southern Asia. Locally, the water extract of *Cleistocalyx operculatus* is commonly used for its anti-hyperglycemic activity [10]. It was reported that DMC promoted glucose uptake and lipid storage in differentiated 3T3-L1 adipocytes [11] and increased glucose uptake and fatty acid oxidation in myotubes [12]. In addition, DMC improved glucose tolerance and significantly increased fatty acid activation (FAO) of the muscles of high-fat diet (HFD)-fed mice [12]. Chalcones can not only be biosynthesized by plants but can also be manufactured synthetically. The simple chalcone structural scaffold can be modified at multiple sites with different chemical moieties. The resulting molecules could respond to different molecular targets or interfere with different signaling pathways [13,14,15]. Using a simple synthetic method, we generated a new artificial chalcone, i.e., 3,5-dimethyl-2,4,6-trimethoxychalcone (TriMetChalc), which was derived from 2′,4′-Dihydroxy-6′-methoxy-3′,5′-dimethylchalcone (DMC). Therefore, the aim of this study was to test the effect of TriMetChalc on food intake and weight gain in lean C57Bl/6J mice and leptin-deficient *ob/ob* mice and to evaluate the impact of this molecule on hepatic steatosis that occurs naturally in *ob/ob* mice.

## 2. Results

### 2.1. Effect of Acute per os Administration of TriMetChalc on Food Intake in C57Bl/6 Mice

Based on DMC structure, we generated here a new artificial chalcone, i.e., 3,5-dimethyl-2,4,6-trimethoxychalcone (TriMetChalc) and evaluated its biological properties on energy metabolism. Both different steps allowing for the synthesis of TriMetChalc and the intermediate compounds are illustrated in Figure 1A. ^1^H proton and ^13^C carbon NMR spectra of the final product are showed in Figure 1B,C, respectively.

A single oral administration of TriMetChalc resulted in a dose-dependent decrease in daily food intake in the C57Bl/6 mice (Figure 2A). Note that 6.5 and 65 mg/kg BW failed to modify food intake measured during the first 24 hours (h) following administration, while 130 and 650 mg/kg BW doses of TriMetChalc diminished it by 11.8 and 18.4%, respectively. Food consumption measured 3, 12, and 24 h after treatment revealed that 130 and 650 mg/kg BW doses of TriMetChalc mainly affected the night-time food intake (Figure 2B). To decipher the feeding behavior analysis after the TriMetChalc administration, we quantified the consumption of a non-nutritive substance, i.e., kaolin. This behavior, known as pica, serves as a model for the study of nausea/emesis in rodents [16]. The effects of the different TriMetChalc doses were compared to those of lithium chloride (150 mg/kg BW), a compound well known to induce nausea and pica behavior in rodents. Mice treated with the vehicle consumed 0.14 ± 0.03 mg/24 h of kaolin, and those treated with lithium ingested 0.33 ± 0.08 mg/24 h (Figure 2C). While the higher dose of TriMetChalc at 650 mg/kg BW caused a significant increase in kaolin intake (0.27 ± 0.03 mg/24 h; *p* < 0.01), the dose of 130 mg/kg BW of TriMetChalc did not induce significant kaolin intake (0.19 ± 0.02 mg/24 h).

### 2.2. Brain Pattern of Central Pathways Activated in Response to Acute TriMetChalc per os Administration

Central structures activated in response to *per os* administration of 130 mg/kg BW TriMetChalc were next identified using the immune detection of the early gene c-Fos. A very low basal number of c-Fos-positive nuclei were observed in the brainstem, pons, and forebrain of the vehicle-treated mice. The TriMetChalc-treated mice showed a strong increase in the number of c-Fos-positive nuclei within the nucleus tractus solitarius (NTS) regardless the rostro-caudal level considered and a moderate increase within the area postrema (AP; Figure 3A,B). The animals challenged with TriMetChalc also displayed a strong rise in c-Fos immunoreactivity in the within forebrain structures such as the arcuate nucleus (ARC), paraventricular hypothalamus nucleus (PVN), and ventromedial hypothalamus (VMH; Figure 3C,D).

### 2.3. Effect of Acute per os Administration of TriMetChalc on Food Intake and c-Fos Expression in ob/ob Mice

The *ob* mutation is a spontaneous recessive mutation in the gene encoding leptin. Mice homozygous for this mutation are leptin-deficient and develop obesity, which appears around 4–5 weeks of age. In this model, obesity is associated with hyperphagia, hyperglycemia, and type II diabetes. To shed light on the mechanisms by which TriMetChalc reduces food intake, we tested the potential interaction between TriMetChalc and endogenous leptin. Thus, we next tested the acute administration of TriMetChalc in *ob/ob* mice. As previously observed in the C57Bl/6 mice, TriMetChalc at 130 and 650 mg/kg BW reduced night-time food intake (Figure 4A,B). The extent of the TriMetChalc-induced food reduction in the C57Bl/6 and *ob/ob* mice was similar (Figure 2 and Figure 4). As previously observed with the C57Bl/6 mice, TriMetChalc increased kaolin intake at a dose of 650 mg/kg BW but not when administered at 130 mg/kg BW (Figure 4C). c-Fos immunochemistry performed in *ob/ob* mice confirmed this observation. The brain pattern and number of c-Fos-positive cells in response to TriMetChalc in the *ob/ob* mice resembled those observed in the C57Bl/6 mice. Indeed, an increase in c-Fos-positive cells was observed in the NTS, the AP, and several nuclei of the hypothalamus, i.e., PVN, ARC, and VMH (Figure 5A–D)**.** Other regions appeared virtually devoid of c-Fos-positive nuclei (Figure 5A,B).

### 2.4. Effect of Chronic TriMetChalc Treatment on Food Intake and Body Weight of Leptin-Deficient Mice

We next sought to determine whether the anorexigenic effect of TriMetChalc could reverse leptin-deficiency-induced obesity. To this aim, *ob/ob* mice were chronically administered with TriMetChalc once a day for three weeks. We chose to use TriMetChalc at a dose of 65 mg/kg BW, which appeared as a subthreshold dose in the experiments measuring 24 h food intake over its acute administration. Chronic TriMetChalc administration limited the weight gain of the *ob/ob* mice over the period studied (Figure 6A,B). The weight gain was 7.1 ± 1.4 g for the mice that received the vehicle and 4.0 ± 1.6 g for those treated with TriMetChalc (Figure 6A,B). At the same time, the cumulative food intake over 3 weeks was reduced by 20.9 ± 5.6% following the TriMetChalc administration (Figure 6C,D). Interestingly, the daily food intake was not significantly reduced in the first week but was decreased from day 7 until the end of treatment (Figure 6D), suggesting a cumulative effect of TriMetChalc. We next evaluated the potential impact of chronic TriMetChalc treatment on glucose intolerance, since *ob/ob* mice develop hyperglycemia and a type II diabetes syndrome. OGTT was performed before and after three weeks of treatment with 65 mg/kg BW TriMetChalc. The results confirmed the development of glucose intolerance over time in this model (Figure 6E,F). TriMetChalc significantly reduced the glycemic response to oral glucose overload in the *ob/ob* mice, with a more rapid return to basal glycaemia in the treated group than in the control animals (Figure 6E). Quantification of the area under the curve confirmed this observation (Figure 6F).

### 2.5. Effect of Chronic TriMetChalc Treatment on Hepatic Lipid Content and PLIN2 Expression in ob/ob Mice

After the animal sacrifice, observation of the livers from TriMetChalc-treated mice (65 mg/kg BW) revealed a slightly less yellow coloration, suggesting reduced lipid storage in the test group. Weighing the livers after three weeks of treatment showed that TriMetChalc significantly reduced the weight of this organ (Figure 7A). ORO staining is a commonly used experimental technique to detect the lipid content in cells or tissues. As shown in Figure 7B, ORO staining confirmed lipid liver storage in the *ob/ob* mice and a reduced accumulation in the TriMetChalc-treated animals. Quantification of ORO staining objectified this observation, and a significant reduction in the lipid droplet number and total droplet surface was observed in the TriMetChalc-treated group compared to the control group (Figure 7C,D). The size of the lipid droplets remained unchanged (Figure 7E). Perilipin 2 (PLIN2), also known as adipose-differentiation-related protein (ADRP), was initially thought to be expressed only in adipose tissues. However, it was later found to be expressed in many non-adipose tissues, including the liver, where PLIN2 is associated with lipid droplets and the formation of fatty liver by increasing the uptake of fatty acids [17]. Moreover, a decreased PLIN2 expression has been shown to reduce liver steatosis [18]. PLIN2 immunohistochemistry performed on liver sections of the *ob/ob* mice revealed a strong expression in these animals. This expression was not homogeneous throughout the liver tissue, showing a particularly significant expression in hepatocytes located in the vicinity of the centrolobular vein. In addition to the cytoplasmic location, an accumulation of PLIN2 was also observed around lipid droplets. A clear reduction in PLIN2 expression was observed in the TriMetChalc-treated animals, with an evident reduction in PLIN2 surrounding lipid droplets (Figure 7F).

To enable a thorough understanding of the effect of TriMetChalc on MAFLD, which is known to develop in *ob/ob* models, a lipidomic analysis of the liver was performed [19,20]. The results are expressed in terms of the relative abundance of each compound as the area ratio between the area of the detected species and the area of the internal standard normalized per mg of proteins in the sample. We firstly quantified the liver content of cholesterol and triacylglycerols (TAGs). Despite a clear downward trend towards lower concentrations in terms of both cholesterol and TAG, no significant difference could be demonstrated with the sampling and the administration protocol used (Figure 8A–C and Table 1).

Fatty acids (FAs) constitute the fundamental structural components of complex lipids and can enter the system through dietary intake, and they are either released from visceral adipose tissue during lipolysis or are synthesized within the liver through de novo lipogenesis (DNL). Considering the consistent association between hepatic steatosis and saturated (SFAs), monounsaturated (MUFAs), and polyunsaturated (PUFAs) fatty acids, an investigation of their levels across the studied groups was performed. Analysis of the FA contents within the liver of the control and TriMetChalc (65 mg/kg BW)-treated mice revealed a significant reduction in total FAs (Figure 9A and Table 2). Regarding SFAs, a clear reduction in total SFAs was observed in the *ob/ob* mice that received TriMetChalc (Figure 9B and Table 2). Palmitic acid (FA C16_0) and stearic acid (FA C18_0) were the most prevalent SFAs in the liver samples from the *ob/ob* mice, and their abundance was reduced by the TriMetChalc treatment (Figure 9C,D and Table 2). MUFAs, including palmitoleic acid (FA C16_1), oleic acid (FA C18_1), and eicosenoic acid (FA C20_1), were found to be significantly reduced by the TriMetChalc administration (Figure 9E–H and Table 2). Regarding PUFAs, linoleic acid (FA C18_ω 6) and alpha-linolenic acid (FA C18_3ω3) were found to be increased after the TriMetChalc treatment (Table 2). Accordingly, the total conjugated linoleic acid (CLA-FA) was increased in the TriMetChalc group, although the total PUFAs content was not statistically increased in response to the TriMetChalc treatment (Figure 9I,J and Table 2). Altogether these results showed an increase in the PUFA/SFA and USFA/SFA ratios and a reduction in the MUFA/PUFA ratio (Figure 9K and Table 2). The lipogenic index derived from the ratio of palmitic acid (FA C16_0) to the essential ω-6 linoleic acid (FA C18_2ω6) reflects the rates of DNL. Here, we report a strong reduction in the lipogenic index, as follows: 8.20 ± 0.66 versus 3.24 ± 0.73 in the control and TriMetChalc-treated mice, respectively (Figure 9L and Table 2). We next evaluated the liver phospholipid, ceramide, and sphingomyelin contents in the control and treated groups. Interestingly, phosphatidylcholines (PCs) and phosphatidyl inositols (PIs) were weakened by the treatment, while the levels of phosphatidyl ethanolamines (PEs) and phosphatidyl serines (PSs) were not significantly reduced despite a trend towards lower concentrations (Figure 10A–D and Table 3). As a result, the PCs/PEs ratio was significantly reduced (32.5 ± 0.9 *vs*. 29.2 ± 0.7, *p* < 0.01) in the control and TriMetChalc-treated groups, respectively (Figure 10E). The contents of ceramides and sphingomyelins remained unchanged by the TriMetChalc treatment (Appendix A).

## 3. Discussion

The present study was designed to evaluate the potential therapeutic effects of TriMetChalc on food intake, obesity, and liver steatosis using the hyperphagic, obese, and diabetic *ob/ob* mice model. We observed that TriMetChalc, which was derived from DMC. retained the anti-obesity and anti-diabetic effects of the native molecule, and we also showed that this molecule has anorexigenic properties and confers protective action on the liver, two actions which have not been described for DMC.

### 3.1. TriMetChalc Acts as an Anorexigenic Molecule Independently of Leptin

*Per os* administration of TriMetChalc induced a dose-dependent reduction in daily food intake by decreasing night-time food consumption. This result proves that TriMetChalc was stable and effective in vivo after daily administration. An interesting finding of the present work is that the anorexigenic effect of TriMetChalc was similar in both the C56Bl/6 and leptin-deficient *ob/ob* mice. Leptin is a hormone secreted by adipose cells that inhibits hunger. Leptin-deficient *ob/ob* mice are hyperphagic and develop severe obesity, insulin resistance, and steatosis. Therefore, TriMetChalc acts on leptin-independent signaling pathways or molecular targets operating downstream of leptin. Thus, the anorexigenic action of TriMetChalc in the *ob/ob* mice is of interest in the perspective of the future development of TriMetChalc as an anti-obesity drug, since human obesity is characterized by leptin resistance [21]. A reduction in food intake may be the result of intoxication and the appearance of nausea and vomiting [22]. To verify whether this was the case with TriMetChalc, we measured the consumption of clay (kaolin), used here as an indicator of nausea and emesis [16]. A dose of 130 mg/kg BW, which induced a reduction in food intake, did not lead to any kaolin consumption, excluding the presence of nausea at this dosage. On the other hand, at a high dose, i.e., 650 mg/kg BW, TriMetChalc significantly increased the ingestion of kaolin, showing that a high dosage of this molecule is likely to induce discomfort in animals. In terms of toxicity, we did not observe any remarkable histological liver damage after the administration of TriMetChalc for 3 weeks. In addition, the results of the single-dose acute-toxicity study showed no toxicity or abnormal clinical signs in the mice. The estimated LD 50 is higher than 1300 mg/kg, which is much higher than the effective dosages (6.25 and 650 mg/kg) administered to the mice in this study. Taking these results together, the TriMetChalc compound demonstrated no significant toxicity at the doses used in the present work.

By controlling hunger, the brain plays an essential role in regulating food intake and energy expenditure. Two central regions, i.e., the hypothalamus and dorsal vagal complex (DVC), a brainstem structure, strongly contribute to the homeostatic control of the energy balance by integrating information linked to nutritional status and arising from peripheral organs (gut, liver, pancreas, adipose tissue; see [23] for a review). To decipher the mechanisms underlying the modulation of food intake by TriMetChalc and to identify the central structures activated by this compound, we performed a c-Fos expression mapping. The detection of the early gene c-Fos is classically used to identify brain-activated cells [24,25]. We observed that TriMetChalc strongly and specifically activated hypothalamic nuclei such as ARC, PVN, VMH, and the brainstem nuclei of the DVC. It should be noted that TriMetChalc did not cause c-Fos activation outside these areas, excluding a nonspecific toxic effect. On the contrary, the specific activation of these areas reinforces the idea that TriMetChalc interacts with pathways dedicated to the regulation of food intake. As mentioned previously, TriMetChalc is a derivative of DMC, a major compound in *Cleistocalyx operculatus* extract. Choi and colleagues [12] reported that DMC activated AMP-activated protein kinase (AMPK) by direct binding and proposed that DMC is an AMPK agonist. AMPK was highly expressed in the different hypothalamic nuclei, such as ARC, PVN, and VMH [26]. Hypothalamic AMPK is involved in the modulation of the energy balance and plays an important role in the regulation of feeding. The modulation of the hypothalamic AMPK level and activity by key hormones such as leptin and ghrelin is implicated in the control of food intake and appetite [27]. For example, during fasting, AMPK activity is raised in hypothalamus nuclei, and it is inhibited after refeeding. Furthermore, it has been shown that elevated hypothalamic AMPK activity causes increased food intake leading to weight gain. Conversely, hypophagia and weight loss were caused by its suppression [28]. Thus, given the anorexigenic effect observed here with TriMetChalc, it seems unlikely that it acts as a hypothalamic AMPK agonist. We showed here that TriMetChalc limited the weight gain of the *ob/ob* mice over a period of three weeks, a duration similar to that used in the two previously cited studies [12,29]. Three weeks of DMC treatment improved glucose tolerance in HFD-induced obese mice, but the body weight of DMC-treated mice remained unchanged compared with that of the control group [12]. Recently, Lee and collaborators [29] synthesized 12 chalcone derivatives from DMC possessing AMPK agonist activity. These derivatives increased the fatty acid oxidation (FAO) rate of C2C12 myotubes. After three weeks of administration, the more-potent effective AMPK agonist derivate was reported to improve glucose tolerance in HFD-mice without any modification in weight gain [29]. In addition, we revealed that TriMetChalc reduced food intake from the 7th day until the end of treatment. As a result, the cumulative food intake over the 3-week period was reduced. This is a notable difference from the works of Choi et al. [12] and Lee et al. [29], which suggests that modifying DMC with our method allows the molecule to have an anorexigenic effect. Other synthetic chalcones derivatives, i.e., halogen-containing chalcone derivatives 2-bromo-4′-methoxychalcone and 2-iodo-4′-methoxychalcone, have been reported to prevent body weight gain and impair glucose tolerance in HFD mice after 10 weeks of treatment [30]. But, in the absence of any measurement of food intake in this latest study, it is difficult to compare the effects of these different molecules. To date, our work is the first to show that a chalcone derivative has a hunger-killing effect by directly or indirectly targeting neural networks dedicated to the central energy balance. Understanding the precise mechanism remains to be clarified.

### 3.2. TriMetChalc Attenuates Metabolic-Dysfunction-Associated Fatty Liver Disease

Excess fat deposition within the liver has been reported for centuries. In 1980, Ludwig and colleagues described the liver histology associated with excess liver fat in the absence of significant alcohol consumption and used the term “non-alcoholic steatohepatitis (NASH)” [31]. Non-alcoholic fatty liver disease (NAFLD) is rapidly becoming a worldwide public health problem. NAFLD represents a spectrum of diseases ranging from “simple steatosis”, which is considered relatively benign, to non-alcoholic steatohepatitis (NASH) and NAFLD-associated cirrhosis and end-stage liver disease. Even if NAFLD can be induced by a variety of drugs and toxins, a high prevalence of NAFLD in patients with obesity, metabolic syndrome, and type-2 diabetes has been clearly observed [32]. In 2020, to better reflect the metabolism-related etiology, an international panel of experts reached a consensus to replace the term NAFLD with “metabolic-dysfunction-associated fatty liver disease” (MAFLD) [4,5]. In addition to obesity and type 2 diabetes, *ob/ob* mice develop steatosis and associated lipotoxicity and lipo-apoptosis. Their liver histology shows a rare progression to cirrhosis, since *ob/ob* mice are resistant to hepatic fibrosis. Although *ob/ob* mice do not display the full spectrum of human NASH, these obese animals could represent a suitable model to study MAFLD [19].

Numerous chalcones, i.e., trans-chalcone, naringenin chalcone, xanthohumol, 4-hydroxyderricin, xanthoangelol, cardamonin, flavokawain B, and safflower yellow, have been shown to have protective effects on the liver and in MAFLD. To achieve this, they can improve adipocyte functions, increase adiponectin secretion, inhibit lipogenesis, or enhance fatty acid oxidation (see [9] for a review). However, no such effects have been reported for DMC. Only one study has shown that DMC has a hepatoprotective effect on carbon-tetrachloride-induced acute liver injury in Kunming mice through the attenuation of oxidative stress and inhibition of lipid peroxidation [33]. Here, we observed that the treatment for three weeks with a dose of 65 mg/kg BW of TriMetChalc significantly reduced hepatic steatosis in the *ob/ob* mice, as quantified by histological staining and PLIN2 immunohistochemistry. This is the first observation to show that a DMC derivative compound can have a positive impact on this pathology. An important aspect of MAFLD progression is lipotoxicity, which is caused by the hepatic accumulation of lipids that can trigger cellular stress responses. We performed a lipidomic study to identify which lipids were modified by the treatment. Despite a trend towards a reduction in TAG and cholesterol, the quantification carried out here was unable to highlight a significant difference. The three-week exposure time may explain this result, and we will proceed with longer treatment periods in the future. We might also consider increasing the dose of TriMetChalc and/or combining TriMetChalc with other chalcones or compounds with liver-protective action. However, our protocol induced many modifications in long-chain FFAs (containing 16 carbons or more), which were divided into SFAs, MUFAs, and PUFAs based on the presence of double bonds. First, we observed a significant reduction in total the SFAs content, largely due to the decrease in palmitic and stearic acids. Increased levels of hepatic SFAs have been reported in MAFLD patients and animal models, largely due to significant increases in palmitic and stearic acids (see [34] for a review). The excessive accumulation of these two SFAs in hepatocytes has been shown to induce endoplasmic reticulum stress, leading to apoptosis and hepatocyte damage. Hence, palmitic and stearic acids induce concentration- and time-dependent lipo-apoptosis in hepatocytes [35,36]. Palmitic acid, through its interaction with toll-like receptor type 2, has been shown to activate the inflammasome in Kupffer cells and macrophages in vitro and in vivo and then contribute to NASH development [37]. Conversely, an excessive consumption of SFAs may be a risk factor for MAFLD pathogenesis [38].

Animals and humans with MAFLD also have high levels of hepatic total MUFAs, suggesting a potential link between increased levels of these lipids and hepatic inflammation and lipotoxicity [39,40,41]. The most abundant and well-studied MUFAs in MAFLD are palmitoleic acid (C16:1) and oleic acid (C18:1), which are generated from the SFAs palmitic and stearic acids, respectively [42]. Oleic-acid-induced steatosis has been reported in hepatocytes of many mammalian species [43,44]. Although they contribute to steatosis, these MUFAs are less lipotoxic than SFAs. Indeed, individually studied, these MUFAs induce apoptosis, but this effect is minimal compared with SFAs [42]. Furthermore, they have been shown to greatly attenuate palmitate-induced apoptosis in cultured hepatocytes [45]. Consistently, in cultured hepatocytes, MUFAs have been shown to reduce palmitate-induced apoptosis but increase the accumulation of triglycerides (TAGs) [46]. Moreover, in the context of impaired TAGs synthesis, oleic acid accumulation exerts significant lipotoxicity [46]. Finally, the inhibition of stearoyl-CoA desaturase 1 (SCD1), a rate-limiting enzyme that allows for the biosynthesis of monounsaturated fatty acids from their saturated fatty acid precursors, has been proposed as a treatment for MAFLD [47,48]. In this context, the concomitant decrease in SFAs and MUFAs, i.e., FA C16:0, FA C16:1, FA C18:0, FA C18:1, and FA C20:1, observed in the TriMetChalc-treated *ob/ob* mice could help to reduce lipotoxicity and TAG accumulation. Our lipidomic analyses revealed a significant increase in linoleic acid (FA C18_2ω6) and alpha-linolenic acid (FA C18_3ω3). However, this increase, limited to these two PUFA species, did not translate into an overall increase in PUFA contents. However, considering the decreased level of SFAs after TriMetChalc administration, the PUFAs/SFAs ratio was found to be increased by the TriMetChalc administration. In recent years, it has become clear that PUFAs are important for many biological processes within cells. For instance, PUFAs increase membrane fluidity, which raises the number of membrane insulin receptors and improves insulin sensitivity. Additionally, through their interactions with other genes, PUFAs directly regulate the activity of genes linked to lipid metabolism, redox balance, inflammation, and fibrogenesis in NASH (see [42,49] for reviews). The PUFAs content is altered in MAFLD, and the severity of this pathology is correlated with a constant decline in hepatic PUFA levels [39,50]. Thus, the molar percentages of both ω-3 and ω-6 PUFAs were decreased in human liver biopsies from MAFLD patients [39]. A low liver PUFAs/SFAs ratio has been associated with an increased risk of developing atherosclerosis, cardiovascular diseases, diabetes, and MAFLD [51]. In a meta-analysis, Yu and colleagues [52] found that total PUFAs improve liver function and promote benefits in obesity-related comorbidities, such as a reduction in insulin resistance, dyslipidemias, inflammation, and non-alcoholic fatty liver disease markers. PUFAs are effective at limiting the hepatic steatosis process by increasing the gene expression of lipid oxidation, reducing the activity of liver lipogenic enzymes, and releasing adiponectin [53]. Although the overall PUFA concentration was not modified by the TriMetChalc treatment, the increase in linoleic and alpha-linolenic acid PUFAs and the concomitant reduction in SFAs and MUFAs induced by TriMetChalc should be highlighted and could contribute to improving the liver status of these animals. Liver steatosis in MAFLD is triggered by excessive hepatic triglyceride synthesis using white adipose tissue (WAT)-derived fatty acids, de novo lipogenesis (DNL), and endocytosis of triglyceride-rich lipoproteins. DNL, the metabolic pathway synthesizing SFAs and MUFAs from acetyl-CoA, accounts for only a small fraction of FAs in the liver of lean humans (5%) [3]. However, in MAFLD, hepatic DNL is strongly increased, and approximately 25% of triglycerides originate from DNL. This increase in MAFLD compared to healthy subjects can be largely explained by the induction of DNL enzymes [54]. The lipogenic index was introduced by Hudgins et al. in 1996 [55] to reflect the DNL rate and is calculated as the ratio of palmitic acid (16_0) to the essential omega-6 linoleic acid (18_2ω6). The amounts of FA chains (i.e., FA C16_0, FA C18_0, and FA C18_1) representing DNL were substantially elevated in *ob/ob* mice [56]. Here, we reported a clear and strong reduction in the DNL index in response to TriMetChalc treatment, confirming the potential benefit of this molecule in the MAFLD context.

An interesting observation we made is the reduction in hepatic PC contents and the PCs/PEs ratio with TriMetChalc treatment. PCs and PEs are the two most abundant phospholipids in the plasma membranes of all mammalian cells. PCs and PEs predominantly reside on the outer and inner leaflets of the plasma membrane bilayer, respectively. In the context of MAFLD, PCs and PEs have been associated with liver damage. In several mouse models and in human studies, a change in not only the absolute concentrations of these phospholipids but, more critically, in the molar ratio between PCs and PEs is a key determinant of liver health [57]. Mostly, the severity of MAFLD is generally negatively correlated with the hepatic PCs/PEs ratio [58]. But, in some models, the PCs/PEs ratio can be increased above the normal range, and this disturbance in the ratio is also associated with MAFLD. Leptin-deficient *ob/ob* mice belong to models where the hepatic PCs/PEs ratio is increased [59]. To manage excess lipids storage in these mice, the expression of genes involved in hepatic PC synthesis (i.e., Pcyt1a and Pemt) is up-regulated, increasing the hepatic PCs/PEs ratio [60]. When *ob/ob* mice were treated with Pemt-targeted shRNA, the hepatic PCs/PEs molar ratio was restored to the wild-type level, thereby significantly decreasing hepatic steatosis [57]. In this context, the reduction in the PCs/PEs ratio observed in response to the TriMetChalc treatment could contribute to a reduction in liver steatosis in this model and could be an indicator of an improvement in the metabolic status of the *ob/ob* mice.

## 4. Materials and Methods

### 4.1. Animal Housing

Experiments were performed on 7–8-week-old male C57Bl/6J mice and homozygous male C57BL/6J-*lep*^ob^ (*ob/ob*) mice (Charles River Laboratories, l’Abresle, France). All animals were individually housed in a pathogen-free facility at a controlled temperature (21 °C) with a 12/12 h light/dark cycle (lights on at 7 AM) with a standard diet (AO4 P2.5, UAR SAFE, Augy, France) and water available ad libitum. For all experiments, the vehicle- and TriMetChalc-treated groups each included 8 animals, with the exception of the PLIN2 immunochemistry tests, which were performed on groups containing 5 animals. When different doses of TriMetChalc were tested, each dose was evaluated on a group comprising 8 animals.

### 4.2. 3,5-Dimethyl-2,4,6-trimethoxychalcone (TriMetChalc) Synthesis Pathway

The different steps allowing for the synthesis of TriMetChalc are summarized below, and the intermediate compounds are illustrated in Figure 1A. ^1^H proton and ^13^C carbon NMR spectra of the final product are showed in Figure 1B,C, respectively.

#### 4.2.1. Synthesis of 2,4,6-Trimethoxybenzaldehyde

According to a previously described synthesis [61,62], 15 g of 1,3,5-trimethoxybenzene was dissolved in 9 mL of dimethylformamide (1.3 equimolar (eq.)) and cooled to 0 °C. A volume of 9.1 mL of phosphorous oxychloride (1.1 eq.) was slowly added dropwise, and the mixture was allowed to react at room temperature for 1 h. After completion, the mixture was cooled with ice-cold water and neutralized with 8 M KOH. The precipitate was filtered, and a pink solid was obtained (93% yield). ^1^H NMR (300 MHz, DMSO-*d*_6_): δ 10.19 (s, ^1^H, CHO), 6.23 (s, 2H, CH), 3.86 (s, 3H, O-CH_3_), 3.81 (s, 6H, O-CH_3_).

#### 4.2.2. Synthesis of 2,4,6-Trimethoxytoluene

As described in a previous synthesis [61,62], 10 g of 2,4,6-trimethoxybenzaldehyde was dissolved in 30 mL of ethylene glycol. Then, 11.1 mL of hydrazine hydrate (0.5 eq.) and 14 g of KOH (5 eq.) were added. The solution was then heated to 90 °C for 2 h, and once a foam had formed, the mixture was heated to 145 °C for 2 h. When completed, the reaction was poured onto ice-cold water and acidified with 1 M HCl. The precipitate was filtered and washed several times with brine to obtain a white powder (81% yield). ^1^H NMR (300 MHz, DMSO-*d*_6_): δ 6.19 (s, 2H, CH), 3.74, (s, 9H, O-CH_3_), 1.9 (s, 3H, CH_3_).

#### 4.2.3. Synthesis of 3-Methyl-2,4,6-trimethoxybenzaldehyde

As for the previous formulation, 10 g of 2,4,6-trimethoxytoluene was dissolved in 5.5 mL of dimethylformamide (1.3 eq.) and then cooled to 0 °C. Then, 5.6 mL of phosphorous oxychloride (1.1 eq.) was slowly added dropwise, and the mixture was allowed to react at room temperature for 1 h. After completion, the mixture was cooled with ice-cold water and neutralized with 8 M KOH. The precipitate was filtered to obtain a bright yellow precipitate (89% yield). ^1^H NMR (300 MHz, DMSO-*d*_6_): δ 10.18 (s, ^1^H, CHO), 6.53 (s, ^1^H, CH), 3.91 (s, 3H, O-CH_3_), 3.88 (s, 3H, O-CH_3_), 3.68 (s, 3H, O-CH_3_), 1.96 (s, 3H, CH_3_).

#### 4.2.4. Synthesis of 2,4-Dimethyl-1,3,5-trimethoxybenzene

As with the previous reduction, 10 g of 3-methyl-2,4,6-trimethoxybenzaldehyde (0.05 mol) and 10.5 mL of hydrazine hydrate (0.6 eq.) were dissolved in 30 mL of ethylene glycol. When the reactants were perfectly dissolved, 14 g of KOH (5 eq.) was added portion-wise, and then the reaction was heated up to 90 °C for 2 h. When foam had formed, the mixture was heated up again to 145 °C for 2 h. When the reaction was complete, the mixture was poured onto ice-cold water and acidified with 1 M HCl. The precipitate was filtered and washed several times with brine to obtain a yellowish powder (56% yield). ^1^H NMR (300 MHz, DMSO-*d*_6_): δ 6.41 (s, 1H, CH), 3.77 (s, 6H, O-CH_3_), 3.57 (s, 3H, O-CH_3_), 1.98 (s, 6H, CH_3_).

#### 4.2.5. Synthesis of 2-Hydroxy-3,5-dimethyl-4,6-dimethoxyacetophenone

According to a previously described synthesis [63], 10 g of 2,4-dimethyl-1,3,5-trimethoxybenzene was dissolved in 26 mL of acetic anhydride (5.5 eq.). Then, 6.4 mL of boron trifluoride ethyl etherate (2 eq.) was added dropwise to the mixture. When the addition was completed, the mixture was heated up to 90 °C and stirred for 1.5 h, and then the reaction was allowed to stand overnight without stirring. Then, 30 mL of water was added, and the mixture was stirred again for 10 min. Twenty-five milliliters of ethyl acetate was added, and the mixture was extracted three times. The organic layers were combined and washed several times with brine and then dried under pressure to obtain a dark brown, thick oil (74% yield).

^1^H NMR (300 MHz, DMSO-*d*_6_): δ 12.81 (s, 1H, OH), 3.71 (s, 3H, O-CH_3_), 3.69 (s, 3H, O-CH_3_), 2.65 (s, 3H, COCH_3_), 2.09 (s, 3H, CH_3_), 2.03 (s, 3H, CH_3_).

#### 4.2.6. Synthesis of 2-Hydroxy-3,5-dimethyl-4,6-dimethoxychalcone

A total of 0.224 g of 2-hydroxy-3,5-dimethyl-4,6-dimethoxyacetophenone, 0.106 g of benzaldehyde (1 eq.), and 0.168 g of lithium hydroxide (5 eq.) were added to 10 mL of ethanol. The mixture was then stirred at room temperature for 16 h. Then, the ethanol was evaporated under pressure, and the residual oil was washed with water and 1 M HCl. The precipitate was filtered and then dissolved in ethyl acetate. The mixture was extracted several times with brine, and the ethyl acetate was evaporated to obtain a bright orange oil (89% yield). ^1^H NMR (300 MHz, CDCl_3_): δ 7.42–7.51 (m, 6H), 5.42 (d, 3Hz, 1H), 3.83 (s, 3H, O-CH_3_), 3.75 (s, 3H, O-CH_3_), 2.17 (s, 6H, 2xCH_3_). ^13^C NMR (75 MHz, CDCl_3_): δ 190.2, 163.4, 159.6, 139.1, 128.7, 125.7, 118.6, 115.6, 111.7, 61.1, 60.0, 8.5, 9.0.

#### 4.2.7. Synthesis of 3,5-Dimethyl-2,4,6-dimethoxychalcone (TriMetChalc)

A total of 200 mg of 2-hydroxy-3,5-dimethyl-4,6-dimethoxychalcone and 265 mg of potassium carbonate (3 eq.) were dissolved in 10 mL of acetone. A volume of 0.121 mL of dimethyl sulfate (2 eq.) was added dropwise, and the mixture was stirred and refluxed for 12 h. When the reaction was complete, the precipitate was filtered, and the solvent was evaporated. The residual oil was dissolved in dichloromethane and extracted several times with brine. The organic layer was evaporated to obtain a bright yellow oil. (76% yield). ^1^H NMR (300 MHz; CDCl_3_): δ 7.50 (m, 2H), 7.29 (m, 4H), 6.95 (d, J = 15 Hz), 3.65 (s, 3H, O-CH_3_), 3.59 (s, 6H, 2xO-CH_3_), 2.13 (s, 6H, 2xCH_3_). ^13^C NMR (75 MHz; CDCl_3_): δ 193.3, 162.0, 159.1, 158.9, 142.8, 135.4, 130.1, 128.9, 128.4, 126.7, 109.1, 108.8, 106.5, 62.3, 8.2, 7.5.

### 4.3. Per os Administration of TriMetChalc

Once a day (i.e., 10 h AM), the mice were administered from 6.25 to 25 mg/kg BW TriMetChalc dissolved in vegetal oil via gavage using a 22-gauge intubation needle (Popper and Sons Inc. Laboratory, New York, NY, USA). Prior to the TriMetChalc treatment, the mice received the same volume of distilled water using a similar oral administration procedure for a habituation period of seven consecutive days.

### 4.4. Food Intake Measurements and Pica Behavior (Kaolin Intake)

Food consumption: Immediately after treatment, a new supply of pre-weighed food was given. Food intake was calculated as the difference between the pre-weighed food and the remaining chow measured with a precision balance (0.01 g; Denver Instrument from Bioblock).

Measurement of pica behavior (kaolin intake): Pica behavior was assessed as previously described [16]. Briefly, kaolin pellets were prepared from pharmacological-grade kaolin and gum Arabic (Sigma Chemical Co) mixed at a 99:1 ratio in distilled water. The kaolin paste was rolled and cut into pieces similar in shape to mice chow pellets. The pellets were dried in an oven at 37 °C for 72 h and then placed in individual cages. The mice were allowed access to regular food and kaolin pellets for a 5-day adaptation period before the beginning of the study. On the day of the experiment, the last 24 h, the kaolin consumption was recorded. The mice were then *per os* administered TriMetChalc (65 mg/kg BW) or a vehicle, as described above, and the kaolin intake was measured 24 h later by subtracting the pre-weighed and remaining kaolin pellets with a precision balance (Denver Instrument from Bioblock, Illkirch, France).

### 4.5. Tissue Histology and Oil Red O Staining

Oil Red O (ORO) staining of hepatic sections was performed as previously described [64]. Briefly, 10 µm thick sections were prepared from fixed liver samples with a cryostat (Leica CM3050, France). An ORO stock solution at 0.625% was first prepared in isopropanol. Sections were then incubated for 6 min in an ORO working solution (1.5 parts of ORO stock solution to 1 part of distilled water) and then washed three times in distilled water. For microscopic observations, the sections were mounted in Mowiol solution. Four microphotographs (x 20) per animal were acquired using a Nikon Eclipse E600 light microscope coupled to a DXM 1200 Camera and the ACT-1 software (2.7 version). The number of lipid droplets per surface unity (mm^2^), the total droplet surface per surface unity (mm^2^), and the lipid droplet size were counted for each condition using the NIH Image J software (1.54g version).

### 4.6. Immunohistochemistry Procedures

The per-os-treated animals used for the immunostaining procedure were sacrificed 3 h after treatment without free access to food. The mice were anaesthetized using an intraperitoneal injection of ketamine (120 mg/kg BW, Imalgène 1000, Boehringer Ingelheim, Lyon, France) and xylazine (16 mg/kg BW, Rompun 2%, Bayer Santé, Lyon, France). Intracardiac perfusion was achieved in 4% paraformaldehyde (PFA). Brains were post-fixed for 1 h in 4% PFA at room temperature, rinsed in PBS, and then cryoprotected for 24–48 h in 30% sucrose at 4 °C. After freezing the brains in isopentane (−40 °C), coronal sections (40 µm thick) were cut on a cryostat (Leica CM3050, Leica Biosystems, Nanterre, France) and collected serially in PBS (0.1 M; pH 7.4). The brains were cut from the caudal brainstem (Bregma −8.24 mm) to the forebrain (Bregma +0.75 mm).

c-Fos immunohistochemistry was performed on free-floating sections using an anti-c-Fos rabbit antiserum synthesized against the N-terminus of human protein (1:3000, Millipore, Burlington, MA, USA), as previously described [24]. Briefly, the free-floating sections were incubated for 10 min in a solution containing 0.3% H_2_O_2_ in 0.1 M PBS for quenching of endogenous peroxidase activity. The sections were first incubated for 1 h in PBS containing 3% normal goat serum (NGS) and 0.3% Triton X-100 and then overnight at room temperature in PBS containing 3% NGS, 0.3% Triton X-100, and anti-c-Fos antibody. A biotinylated goat anti-rabbit IgG (1:400, Vector Labs) was used as a secondary antibody. After incubation with the avidin–biotin complex (1:200, Vector Labs, Newark, CA, USA), horseradish peroxidase activity was visualized using nickel-enhanced diaminobenzidine (DAB) as the chromogen. The reaction was closely monitored and terminated when the optimal intensity was reached (3–5 min) by washing the sections in distilled water. Non-specific labeling was assessed on alternate slices that were processed identically to those above but in which the primary antibody was omitted. Finally, all sections were mounted on gelatin-coated slides, air-dried, and coverslipped with mounting medium. PLIN2 immunohistochemistry was performed on free-floating liver sections (30 µm thick). The sections were first incubated for 1 h in PBS containing 3% normal goat serum (NGS) and 0.3% Triton X-100 and then overnight at room temperature in the same solution with an added anti-PLIN2 rabbit monoclonal antibody (1/400, ab108323, Abcam, Cambridge, UK). An Alexa Fluor 488-conjugated goat anti-rabbit IgG (1:400, A11034, Invitrogen, Illkirch, France) was used as the secondary antibody. Sections were mounted on gelatin-coated slides, air-dried, and coverslipped with mounting medium for fluorescence microscope preparation (Mowiol, Sigma-Aldrich, Saint Quentin-Fallavier, France).

### 4.7. Microscopy, Image Analysis, and Cell Count

c-Fos immunostaining was further analyzed by counting positive nuclei on four sections. c-Fos-positive nuclei counting was performed on photomicrographs acquired using a 10x objective equipped with a DXM 1200 Camera (Nikon, Champigny sur Marne, France) coupled with the ACT-1 software (2.7 version). The microscope was set to a specific illumination level, as was the camera exposure time. c-Fos-positive nuclei were then counted on these pictures by computer-assisted morphometry using the NIH image J software (1.54g version). Images were normalized by subtracting the background determined for each nucleus studied. The c-Fos-stained elements were identified by setting a threshold value (140 grey levels above the background on a 0–255 intensity scale). Counts were manually corrected for overlapping cell nuclei that were counted by the software as unique. Software-generated counts of c-Fos-stained profiles were also manually corrected by excluding positive objects whose area did not exceed 10 pixels (image resolution 150 pixels/inch) corresponding to objects with an area equal to or less than to three square micrometers. PLIN2 fluorescent images (z-stack imaging) were acquired on a confocal microscope (Zeiss LSM 710, Ruel Malmaison, France).

### 4.8. Extraction and Analysis of Neutral Lipids and Phospholipids

We performed quantitative analysis of neutral lipids (free cholesterol, cholesterol ester C16, C18, and C20_4, and triacylglycerols C49, C51, C53, C55, C57, and C59) and of the following classes of phospholipids (PLs): ceramides (Cer18:1), phosphatidylethanolamines (PEs), phosphatidylcholines (PCs), sphingomyelins (SMs d18:1), phosphatidylserines (PSs), and phosphatidylinositols (PIs) in the liver samples. Internal calibrations were achieved with stigmasterol, cholesterol ester C17, and TAG 19 for neutral lipids and Cerd18_1/12_0 20 ng, PE 12_0/12_0 180 ng, PC 13_0/13_0 20 ng, SM d18_1/12_0 20 ng, PI 16_0/17_0 30 ng, and PS 12_0/12_0 1600 ng for phospholipids and ceramides. This method allows a relative quantification of the molecules.

For extraction, each frozen liver sample was crushed with a FastPrep ^®^-24 Instrument (MP Biomedical, Illkirch, France) in 1 mL of water/EGTA 5 mM/methanol (1:2, *v*/*v*). After 2 crush cycles (10 m/s, 2X30s), lipids were extracted according to Bligh and Dyer in dichloromethane/water/methanol/2% acetic acid (2.5:2:2.5, *v*/*v*/*v*) [65] in the presence 100 µL internal standards of neutral lipids and 40 µL internal standards of phospholipids. Samples were centrifuged at 2500 revolutions per minute (RPM/min) for 6 min, evaporated to dryness, and then dissolved in 20 µL of ethyl acetate for neutral lipids or 50 µL of methanol for phospholipids. 

For neutral lipids, 1 µL of the lipid extract was analyzed by a gas chromatography flame ionization detector on a GC TRACE 1300 Thermo Electron system (Thermo Fisher Scientific, Illkirch, France) using Zebron ZB-5MS Phenomenex columns (5% polysilarylene, 95% polydimethylsiloxane, 5 m × 0.25 mm id., 0.25 µm film thickness, Phenomenex, Le Pecq, France) [66]. The oven temperature was programmed to increase from 190 °C to 350 °C at a rate of 5 °C/min, and the carrier gas was hydrogen (5 mL/min). The injector and the detector temperatures were at 315 °C and 345 °C, respectively.

For phospholipids, lipid extracts were analyzed by liquid chromatography mass spectrometry using an Agilent 1290 UPLC system coupled with a G6460 triple quadripole mass spectrometer (Agilent Technologies, Les Ulis, France) and using the MassHunter software for data acquisition and analysis (B.06.00, Service Pack 1). A Kinetex HILIC column (50 × 4.6 mm, 2.6 µm, Phenomenex, Le Pecq, France) was used for the liquid chromatography separations. The column temperature was controlled at 40 °C. The mobile phase A was acetonitrile, and phase B was 10 mM ammonium formate in water at pH 3.2. For Cer, PEs, PCs, and SMs, the gradient was as follows: from 10% to 30% B in 10 min; 10–12 min, 100% B; and then back to 10% B at 13 min for 2 min prior to the next injection. The flow rate of the mobile phase was 0.3 mL/min, and the injection volume was 2 µL. For PIs and PSs, the gradient was as follows: from 5% to 50% B in 10 min and then back to 5% B at 10.2 min for 9 min prior to the next injection. The flow rate of the mobile phase was 0.8 mL/min, and the injection volume was 5 µL. Electrospray ionization was performed in the positive mode for analysis of Cer, PEs, PCs, and SMs and in the negative mode for analysis of PIs and PSs. The needle voltages were set at 4 kV and −3.5 kV, respectively. Analyses were performed in the Selected Reaction Monitoring detection mode (SRM) using nitrogen as the collision gas. The ion optics and collision energy were optimized for each lipid class. Finally, peak detection, integration, and quantitative analysis were carried out using the MassHunter Quantitative analysis software (B.06.00, Service Pack 1, Agilent Technologies, Les Ulis, France).

### 4.9. Extraction and Analysis of Fatty Acids

Quantitative analysis of total conventional fatty acids (FAs, C10_0, C12_0, C14_0, C15_0, C16_0, C17_0, C18_0, C20_0, C22_0, C23_0, C24_0, C14_1ω5, C15_1, C16_1ω7, C18_1ω9, C18_1ω7, C20_1ω9, C22:1ω9, C24_1ω9, C18_2ω6, C18_3ω6, C18_3ω3, C20_2ω6, C20_3ω3, C20_3ω6, C20_4ω6, C20_5ω3, C22_2ω6, C22_6ω3, C22_4ω6) was achieved using either TAG17, TAG19, or TAG15 as the internal standard. This method allowed for the relative quantification of esterified and free fatty acids. Lipids were extracted according to Bligh and Dyer ([65] in dichloromethane/methanol/water (2.5:2.5:2, *v*/*v*/*v*) in the presence of the internal standard glyceryl trinonadecanoate (4 µg). The lipid extract was hydrolyzed in KOH (0.5 M in methanol) at 55 °C for 30 min and transmethylated in 14% boron trifluoride methanol solution (1 mL, Sigma-Aldrich, Saint Quentin-Fallavier, France) and heptane (1 mL) at 80 °C for 1 h. After addition of water (1 mL) to the crude, FAs were extracted with heptane (3 mL), evaporated to dryness, and dissolved in ethyl acetate (20 µL). FAs were analyzed by gas chromatography equipped with an FID on a Clarus 600 Perkin Elmer system (Perkin Elmer, Villebon-sur-Yvette, France) using Famewax RESTEK fused silica capillary columns (30 m × 0.32 mm i.d, 0.25 µm film thickness, Restek, Lisses, France) [67]. The oven temperature was programmed to increase from 100 °C to 250 °C at a rate of 6 °C/min, and the carrier gas was hydrogen (1.5 mL/min). The injector and the detector temperatures were at 220 °C and 230 °C, respectively.

### 4.10. Statistical Analysis

All results are presented as mean ± SEM. Statistical analyses were performed using Graphpad Prism 6.05 (GraphPad Software, La Jolla, CA, USA). Comparisons between two groups were performed using Student’s unpaired 2-tailed *t* test. Pearson’s correlation analysis was used to quantify relationships between variables of interest. Significant differences were assessed by one-way ANOVA followed by Fisher’s *post hoc* test for comparisons between four or five groups (Figure 2, Figure 4, and Figure 6F). Two-way ANOVAs followed by Bonferroni’s multiple comparisons was used to assess effects at different times post treatment (Figure 6A,C). *p*-values less than 0.05 were considered significant.

## 5. Conclusions

In summary, the present work provides the first demonstration of a chalcone derivative capable of reducing both food intake and hepatic steatosis in a mouse model of obesity and MAFLD. Given that in obese subjects, insulin-resistant hypertrophic white adipocytes exhibit high elevated lipolysis, which explains the link between obesity and MAFLD, we cannot exclude the fact that the reduction in food intake and overweight induced by TriMetChalc partly explains the amelioration of MAFLD. Although promising, these results need to be supplemented with additional studies that clarify its mode of action and evaluate its action over longer treatment periods and/or in combination with other chalcones with protective effects on the liver.

## Figures and Tables

**Figure 1 ijms-25-09838-f001:**
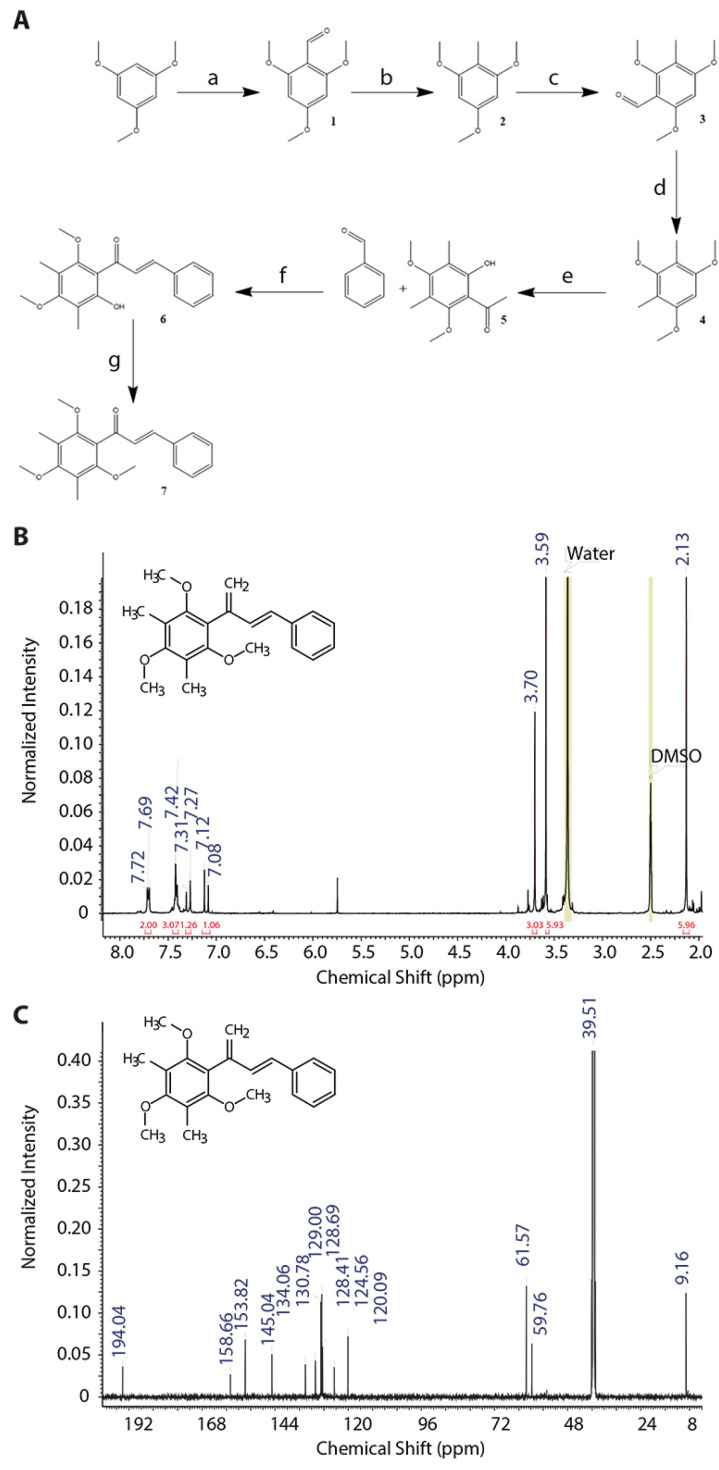
**Synthesis path to obtain 3,5-dimethyl-2,4,6-trimethoxychalcone (TriMetChalc).** (**A**) **a**: Dimethylformamide, phosphorous oxychloride, 0 °C, 1 h. **b**: hydrazine hydrate, KOH, ethylene glycol 90–145 °C, 4 h. **c**: dimethylformamide, phosphorous oxychloride, 0 °C, 1 h. **d**: hydrazine hydrate, KOH, ethylene glycol 90–145 °C, 4 h. **e**: boron trifluoride ethyl etherate, acetic anhydride, 90 °C, 1.5 h. **f**: LiOH, ethanol, 25 °C, 16 h. **g**: K_2_CO_3_, dimethyl sulfate, acetone, 50 °C, 12 h. ^1^H proton (**B**) and ^13^C carbon (**C**) NMR spectra of the final product.

**Figure 2 ijms-25-09838-f002:**
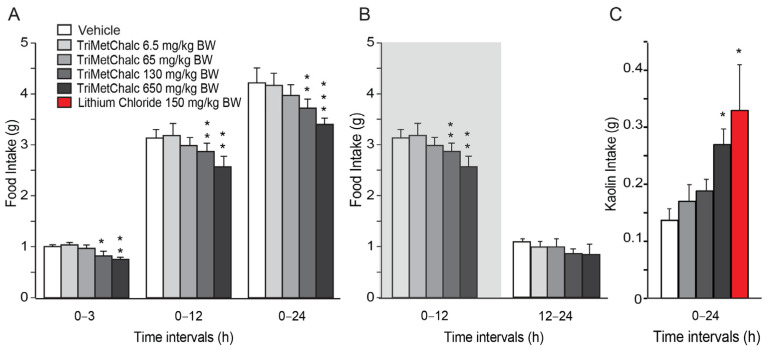
**TriMetChalc reduced food intake.** (**A**,**B**) Cumulative food intake measured over a 24 h period in C57Bl/6J mice *per os* administered with vehicle (*n* = 8) or TriMetChalc (6.5 to 650 mg/kg BW, *n* = 8 for each dose tested). In (**B**), the dark period is represented by a shaded box. * *p* < 0.05, ** *p* < 0.01, *** *p* < 0.001, significantly different from vehicle-treated mice. (**C**) Kaolin intake measured in C57Bl/6 mice 24 h after vehicle, TriMetChalc (65 to 650 mg/kg BW), or LiCl (150 mg/kg BW) administration. * *p* < 0.05 indicates significant difference from vehicle-treated mice.

**Figure 3 ijms-25-09838-f003:**
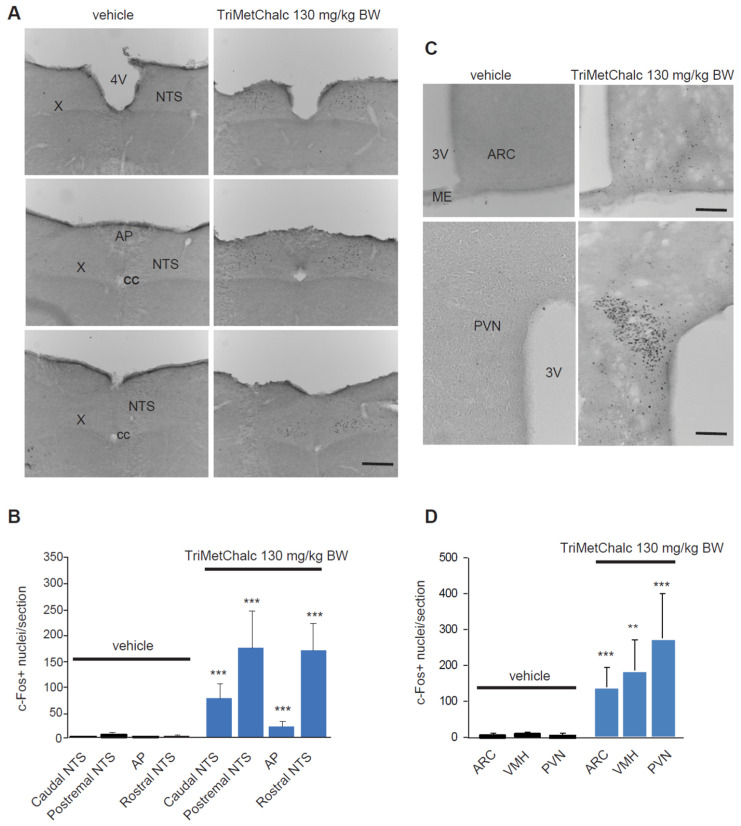
**Identification of TriMetChalc-sensitive brain structures in C57Bl/6J mice.** (**A**–**C**) Representative coronal sections illustrating the c-Fos labeling observed within brainstem (**A**) and hypothalamic (**C**) regions of C57Bl/6J mice treated with vehicle (left panel, *n* = 8) or animals sacrificed 3 h (right panel) after *per os* treatment with TriMetChalc (130 mg/kg BW, *n* = 8). Scale bar: 200 µm. (**B**–**D**) Quantification of the number of c-Fos-immunoreactive nuclei observed within the brainstem (**B**) and hypothalamic (**D**) structures 3 h after treatment with either vehicle or TriMetChalc (130 mg/kg BW). ** *p* < 0.01, *** *p* < 0.001 indicate significant differences from vehicle-treated mice. AP, area postrema; ARC, arcuate nucleus; cc, central canal; ME, median eminence; NTS, nucleus tractus solitarii; PVN, paraventricular nucleus; VMH: ventromedial hypothalamus; X, dorsal motor nucleus of the vagus nerve; 3V, third ventricle; 4V, fourth ventricle.

**Figure 4 ijms-25-09838-f004:**
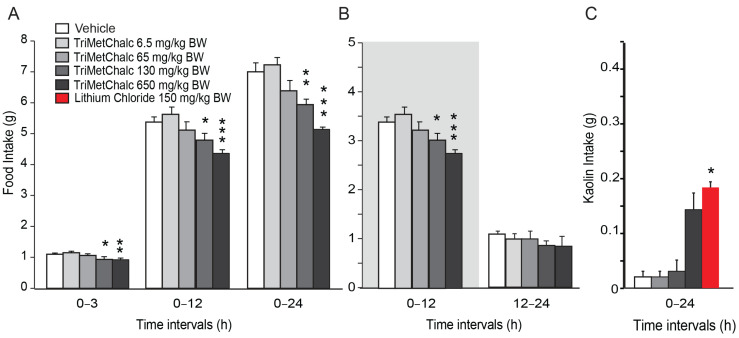
**TriMetChalc reduced food intake in *ob/ob* mice.** (**A**,**B**) Cumulative food intake measured over a 24 h period in *ob/ob* mice administered *per os* with vehicle (*n* = 8) or TriMetChalc (6.5 to 650 mg/kg BW, *n* = 8 for each dose tested). In (**B**), the dark period is represented by a shaded box. * *p* < 0.05, ** *p* < 0.01, *** *p* < 0.001 indicate significant differences from vehicle-treated mice. (**C**) Kaolin intake measured in *ob/ob* mice 24 h after vehicle, TriMetChalc (65 to 650 mg/kg BW) or LiCl (150 mg/kg BW) administration. * *p* < 0.05 significant difference from vehicle-treated mice.

**Figure 5 ijms-25-09838-f005:**
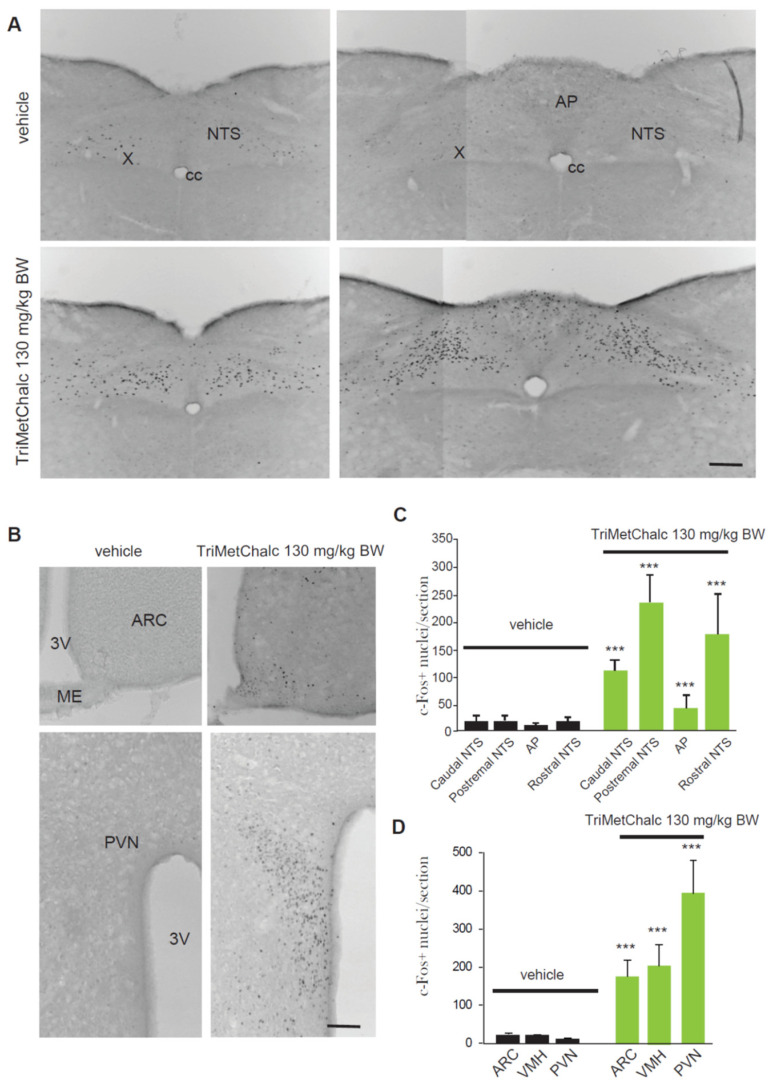
**Identification of TriMetChalc-sensitive brain structures in *ob/ob* mice.** (**A**,**B**) Representative coronal sections illustrating the c-Fos labeling observed within brainstem (**A**) and hypothalamic (**B**) regions of *ob/ob* mice treated with vehicle (*n* = 8) or animals sacrificed 3 h after treatment with TriMetChalc (130 mg/kg BW, *n* = 8). Scale bar: 200 µm. (**C**,**D**) Quantification of the number of c-Fos-immunoreactive nuclei observed within the brainstem (**C**) and hypothalamic (**D**) structures 3 h after treatment with either vehicle or TriMetChalc (130 mg/kg BW). *** *p* < 0.001 indicates significant difference from vehicle-treated mice. AP, area postrema; ARC, arcuate nucleus; cc, central canal; ME, median eminence; NTS, nucleus tractus solitarii; PVN, paraventricular nucleus; VMH: ventromedial hypothalamus; X, dorsal motor nucleus of the vagus nerve; 3V, third ventricle.

**Figure 6 ijms-25-09838-f006:**
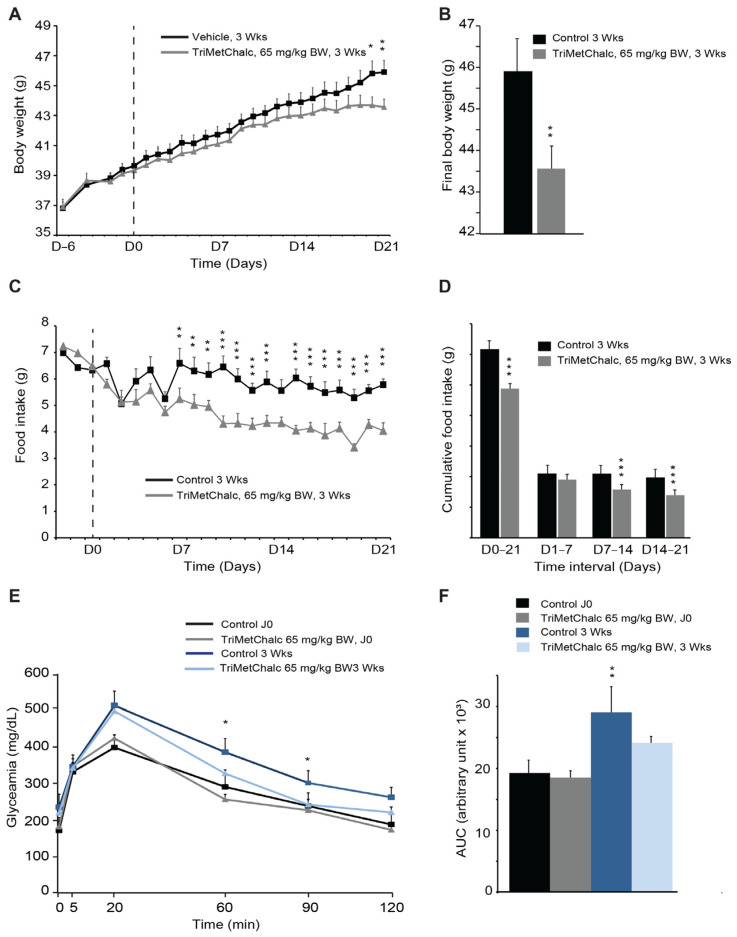
**Chronic TriMetChalc administration reduces leptin-dependent obesity.** (**A**) Body weight measured from day 0 to 21 after *per os* administration of either saline (*n* = 8) or TriMetChalc (65 mg/kg BW, *n* = 8) in *ob/ob* mice. The dashed line indicates the first day of treatment with TrimetChalc or its vehicle. (**B**) Final body weight measured on day 21 after *per os* administration of vehicle (*n* = 8) or TriMetChalc (65 mg/kg BW, *n* = 8) in *ob/ob* mice. * *p* < 0.05, ** *p* < 0.01 indicate significant differences from vehicle-treated mice. (**C**,**D**) Daily food intake (**C**) and cumulative food intake (**D**) measured from day 0 to 21 following administration of either vehicle (*n* = 8) or TriMetChalc (65 mg/kg BW, *n* = 8) in *ob/ob* mice ** *p* < 0.01, *** *p* < 0.001 indicate significant differences from vehicle-treated mice. (**E**) Oral glucose tolerance test using *per os* glucose administration (1.5 g/kg) performed before and after a 21-day period of vehicle or TriMetChalc (65 mg/kg BW) administration in *ob/ob* mice. * *p* < 0.05 indicate significant differences from vehicle-treated mice (**F**) Quantification of area under the curves (AUC) measured at 120 min after *per os* glucose administration. ** *p* < 0.01 indicate significant differences from vehicle-treated mice.

**Figure 7 ijms-25-09838-f007:**
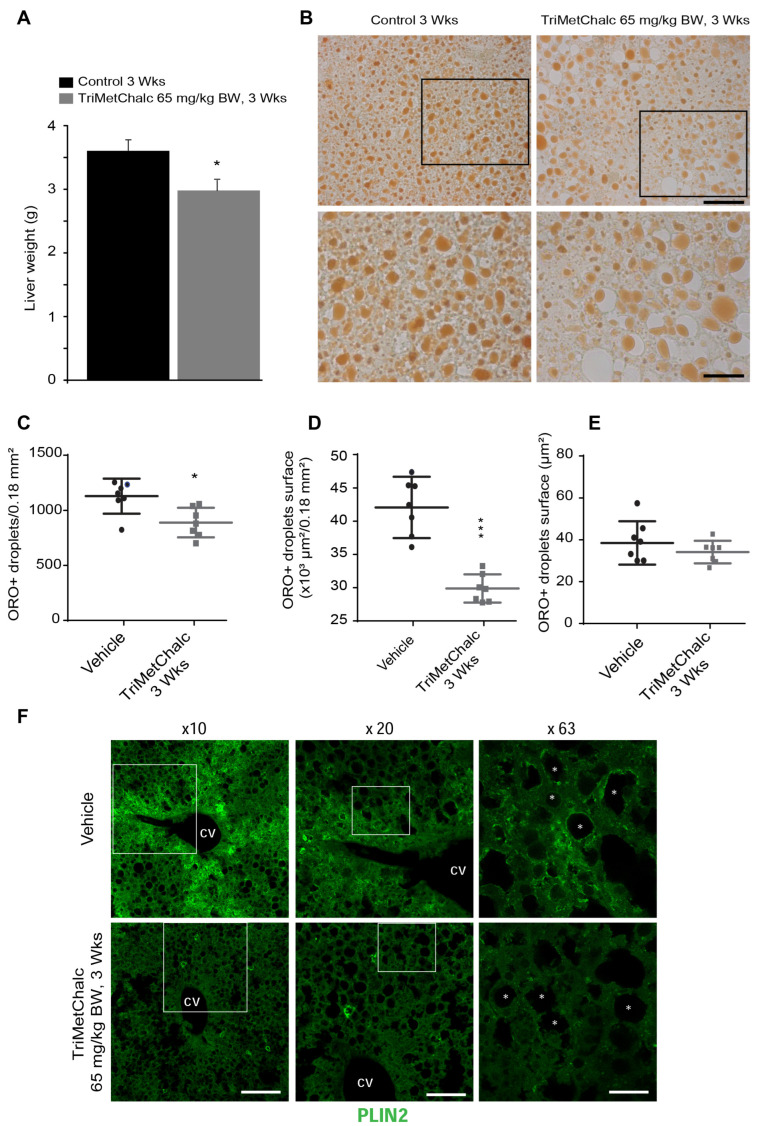
**Chronic TriMetChalc administration ameliorates liver steatosis in *ob/ob* mice.** (**A**) Liver weight measured in control (*n* = 8) and TriMetChalc (65 mg/kg BW, *n* = 8) groups at the end of the treatment period. * *p* < 0.05 indicate significant differences from vehicle-treated mice. (**B**) Representative photomicrographs of ORO staining observed in vehicle- and TriMetChalc (65 mg/kg BW)-treated *ob/ob* mice. Black boxes (upper panels) indicate where high-magnification images (lower panels) originated (Scale bar: 10 µm). (**C**–**E**) Quantification of ORO+ inclusion number per surface unit (**C**), total surface covered by ORO+ inclusions (**D**), and mean surface of ORO+ inclusion (**E**) in liver of vehicle- and TriMetChalc (65 mg/kg BW)-treated mice. * *p* < 0.05, *** *p* < 0.001 indicate significant differences from vehicle-treated mice. (**F**) Representative confocal images of PLIN2 immunohistochemistry performed on liver slices from vehicle (*n* = 5)- and TriMetChalc (65 mg/kg BW, *n* = 5)-treated mice. CV: centrolobular vein. White boxes indicate where high-magnification images originated, asterisks indicate lipid droplets. Scale bars: 200, 100, and 30 µm in ×10, ×20, and ×63 images, respectively.

**Figure 8 ijms-25-09838-f008:**
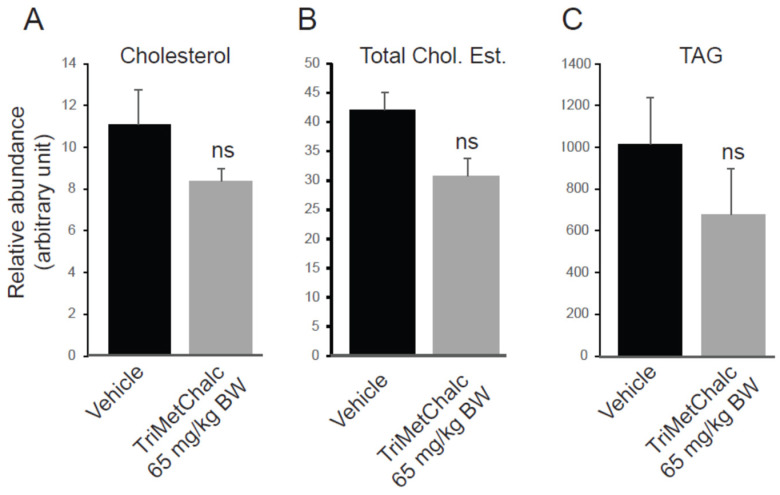
Quantification of free cholesterol (**A**), cholesterol ester (**B**), and triacylglycerols (**C**) in liver extracts from vehicle (*n* = 8)- or TriMetChalc (65 mg/kg BW, *n* = 8)-treated *ob/ob* mice. ns: non-significant difference.

**Figure 9 ijms-25-09838-f009:**
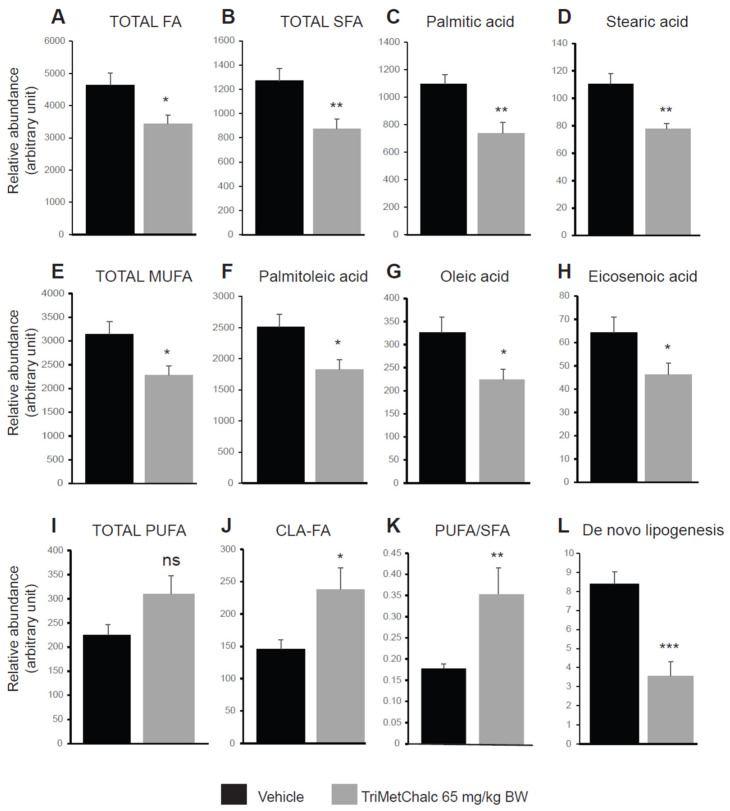
Quantification of FAs (**A**), SFAs (**B**–**D**), MUFAs (**E**–**H**), and PUFAs (**I**,**J**) in liver extracts from vehicle (*n* = 8)- or TriMetChalc (65 mg/kg BW, *n* = 8)-treated *ob/ob* mice. PUFAs/SFAs ratio (**K**) and de novo lipogenesis (**L**) were calculated from data detailed within Table 2. * *p* < 0.05, ** *p* < 0.01, *** *p* < 0.001 indicate significant differences from vehicle-treated mice. ns: non-significant difference.

**Figure 10 ijms-25-09838-f010:**
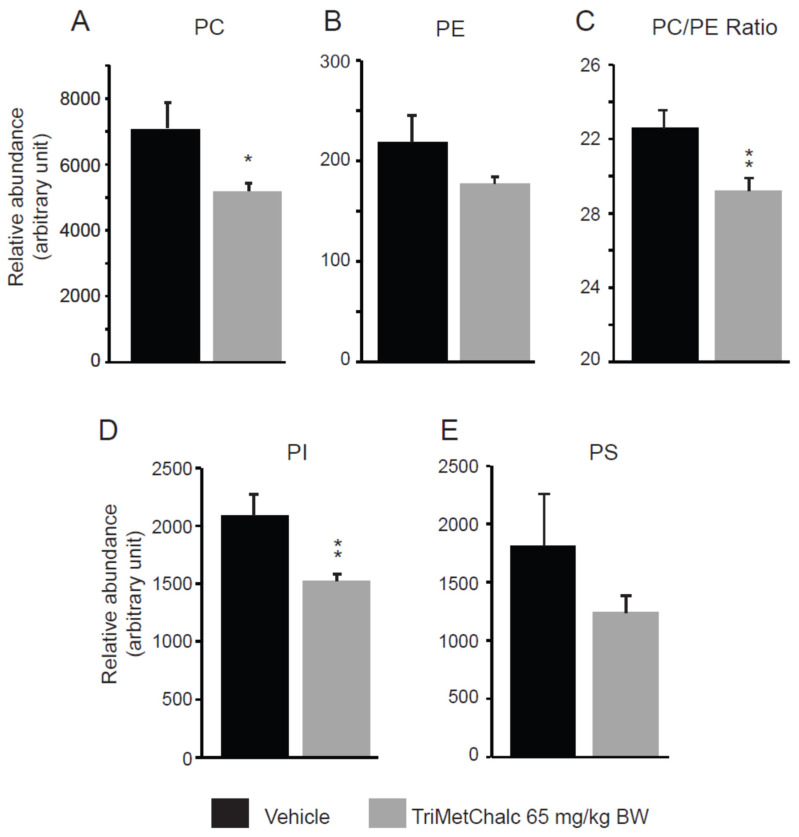
Quantification of phosphatidylcholines (PCs, **A**), phosphatidylethanolamines (PEs, **B**), PCs/PEs ratio (**C**), phosphatidylinositols (PIs, **D**), and phosphatidylserines (PSs, **E**) in liver extracts from vehicle (*n* = 8)- or TriMetChalc (65 mg/kg BW, *n* = 8)-treated *ob/ob* mice. * *p* < 0.05, ** *p* < 0.01 indicate significant differences from vehicle-treated mice.

**Table 1 ijms-25-09838-t001:** Detailed cholesterol and triacylglycerol compositions and quantification of liver extracts from vehicle (*n* = 8)- or TriMetChalc (65 mg/kg BW, *n* = 8)-treated *ob/ob* mice. Statistical differences were examined by Student’s paired *t*-test. *p* values are reported in the last column. Only a *p* value < 0.05 denotes statistical significance between vehicle- and TriMetChalc-treated mice. nd = not detectable.

	Control		TriMetChalc		Control *vs*.TriMetChalc
	Mean	SEM	Mean	SEM	*t*-test
**L_Cholesterol**	11.08	1.67	8.37	0.59	0.15
L_Chol_C16	14.86	2.25	10.84	1.14	0.14
L_Chol_C18	25.45	3.88	18.66	1.79	0.14
L_Chol_C20	1.74	0.16	1.21	0.20	0.07
**Total Chol Est**	42.05	6.18	30.72	3.03	0.13
L_TAG46	nd		nd		
L_TAG48	37.36	12.61	25.17	11.45	0.49
L_TAG50	228.96	83.53	154.07	63.95	0.49
L_TAG52	535.88	200.02	361.51	120.48	0.47
L_TAG54	191.14	79.43	119.89	25.80	0.41
L_TAG56	22.58	7.56	14.68	2.03	0.33
**Total TAG**	1015.94	382.54	675.34	223.35	0.45
**LN Totaux**	1069.08	389.9	714.45	221.14	0.44

**Table 2 ijms-25-09838-t002:** Detailed fatty acid (FA) composition and quantification of liver extracts from vehicle (*n* = 8)- or TriMetChalc (65 mg/kg BW, *n* = 8)-treated *ob/ob* mice. Statistical differences were examined by Student’s paired *t*-test. *p* values are reported in the last column. A *p* value < 0.05 denotes statistical significance between vehicle- and TriMetChalc-treated mice1. nd = not detectable. Fatty acids whose concentrations were statistically decreased following treatment with TriMetChalc appear in green in the table, and those where treatment induced a significant increase in their concentration are shown in red. nd = not detectable.

	Control		TriMetChalc		Control vs. TriMetChalc
Fatty Acid	Mean	SEM	Mean	SEM	*t*-test. *p* value
** Total FA **	** 4641.43 **	** 382.72 **	** 3444.58 **	** 264.42 **	** 0.028 **
** Total SAFA **	** 1269.28 **	** 104.54 **	** 871.72 **	** 80.01 **	** 0.006 **
L_C14_0	12.29	1.35	10.44	1.73	0.417
L_C15_0	1.35	0.22	1.10	0.09	0.332
L_C16_0	1098.45	92.91	739.88	76.59	0.014
L_C17_0	2.42	0.39	1.64	0.11	0.083
L_C18_0	110.73	11.19	77.74	3.77	0.019
L_C20_0	1.20	0.12	1.04	0.12	0.406
L_C21_0	nd		nd		
L_C22_0	nd		nd		
L_C23_0	nd		nd		
L_C24_0	42.84	3.67	40.05	5.45	0.680
** Total MUFA **	** 3147.66 **	** 263.23 **	** 2282.99 **	** 189.88 **	** 0.026 **
L_C14_1ω7	0.54	0.09	0.40	0.08	0.300
L_C15_1ω7	nd		nd		
L_C16_1ω9	85.35	7.39	72.25	9.48	0.301
L_C16_1 ω 7	160.64	12.72	110.38	10.01	0.011
L_C17_1ω7	nd		nd		
L_C18:1 ω 9	2508.61	210.06	1827.84	153.57	0.026
L_C18_1 ω 7	326.27	33.31	228.08	32.08	0.030
L_C20_1 ω 9	64.37	6.67	46.36	4.83	0.054
L_C22_1ω9	1.88	0.21	1.52	0.27	0.326
L_C24_1ω9	nd		nd		
**Total PUFA**	**224.50**	**21.88**	**310.39**	**37.52**	**0.084**
L_C16_2ω6	nd		nd		
L_C16_2ω4	nd		nd		
L_C16_3	4.20	0.60	2.75	0.25	0.051
L_C16_4ω3	nd		nd		
L_C18_2 ω 6	141.41	14.77	228.50	23.12	0.026
L_C18_3ω6	1.55	0.14	1.64	0.17	0.692
L_C18_3 ω 3	2.84	0.40	4.95	0.97	0.050
L_C18_4ω3	nd		nd		
L_C20_2 ω 6	8.86	0.96	3.24	0.79	0.001
L_C20_3ω3	nd		nd		
L_C20_3ω6	12.74	1.39	11.23	0.73	0.357
L_C20_4ω6	40.97	3.78	39.92	3.56	0.844
L_C20_5ω3	2.19	0.27	2.45	0.59	0.694
L_C22_2 ω 6	3.09	0.46	1.72	0.37	0.043
L_C22_3ω3	nd		nd		
L_C22_4ω6	2.30	0.30	2.70	0.22	0.307
L_C22_5ω3	4.36	0.91	5.31	0.91	0.478
L_C22_6ω3	nd		nd		
** MUFA/PUFA **	** 14.19 **	** 0.83 **	** 8.61 **	** 1.35 **	** 0.006 **
**MUFA/SAFA**	**2.48**	**0.05**	**2.64**	**0.08**	**0.141**
**PUFA/SAFA**	**0.18**	**0.01**	**0.35**	**0.06**	**0.019**
**USFA/SAFA**	**2.66**	**0.05**	**2.99**	**0.13**	**0.041**
** DNL **	** 8.20 **	** 0.66 **	** 3.24 **	** 0.73 **	** 0.001 **

**Table 3 ijms-25-09838-t003:** **Detailed phospholipids composition and quantification of liver extracts from vehicle (*n* = 8)- or TriMetChalc (65 mg/kg BW, *n* = 8)-treated *ob/ob* mice.** Statistical differences were examined by Student’s paired *t*-test. *p* values are reported in the last column. A *p* value < 0.05 denotes statistical significance between vehicle- and TriMetChalc-treated mice. Phospholipids whose concentrations were statistically decreased following treatment with TriMetChalc appear in green in the table. nd = not detectable.

	Control		TriMetChalc		Control *vs.* TriMetChalc
	Mean	SEM	Mean	SEM	*t*-test, *p* value
** Total PC **	** 7057.94 **	** 757.8 **	** 5187.62 **	** 281.29 **	** 0.04 **
L_PC28_0	0.09	0.01	0.06	0.005	0.04
L_PC30_0	7.32	0.63	6.01	0.59	0.16
L_PC30_1	0.77	0.08	0.5	0.06	0.03
L_PC32_0	279.32	18.77	249.1	25.07	0.35
L_PC32_1	123.51	17.61	76.98	10.96	0.04
L_PC32_2	8.08	0.88	5.52	0.63	0.04
L_PC34_0	187.36	18.73	125.08	12.102	0.01
L_PC34_1	1673.29	187.02	1065.24	111.26	0.01
L_PC34_2	507.53	64.06	468.49	29.59	0.59
L_PC34_3	28.72	2.31	26.75	1.49	0.49
L_PC36_1	525.28	50.46	294.9	38.24	0.001
L_PC36_2	693.58	81.86	514.46	20.31	0.04
L_PC36_3	487.84	54.12	369.98	14.4	0.06
L_PC36_4	490.33	50.25	475.27	28.86	0.8
L_PC38_2	69.62	8.11	37.29	4.99	0.001
L_PC38_3	104.89	13.74	54.43	7.37	0.001
L_PC38_4	579.10	64.85	484.69	36.01	0.23
L_PC38_5	318.35	27.7	258.8	11.99	0.07
L_PC38_6	574.35	68.66	418.17	14.18	0.04
L_PC40_3	3.29	0.43	2.03	0.24	0.02
L_PC40_6	395.25	58.66	253.76	18.78	0.04
**Total PE**	**218.77**	**26.68**	**177.54**	**8.19**	**0.17**
L_PE32_0	0.51	0.05	0.39	0.01	0.15
L_PE32_1	0.55	0.06	0.34	0.05	0.03
L_PE32_2	0.03	0,00	0.04	0,00	0.23
L_PE34_0	7.67	0.87	5.77	0.51	0.08
L_PE34_1	4.03	0.60	3.57	0.14	0.47
L_PE36_1	11.03	1.32	8.18	0.38	0.06
L_PE36_2	11.59	1.70	9.07	0.35	0.17
L_PE36_3	11.47	1.60	9.70	0.40	0.30
L_PE36_4	11.08	1.36	9.62	0.56	0.34
L_PE38_2	4.05	0.5	2.80	0.17	0.04
L_PE38_3	28,00	3.39	24.3	1.30	0.33
L_PE38_4	37.77	3.93	34.615	1.76	0.48
L_PE38_5	22.67	2.93	16.21	1.00	0.06
L_PE38_6	23.28	2.86	18.88	1.02	0.17
L_PE40_3	0.67	0.06	0.64	0.04	0.68
L_PE40_5	14.94	2.25	10.3	0.59	0.07
L_PE40_6	21.71	2.85	15.86	0.85	0.07
L_PE40_7	7.64	0.67	7.179	0.32	0.53
** Total PI **	** 2088.61 **	** 184.57 **	** 1519.7 **	** 52.79 **	** 0.01 **
L_PI32_1	1.96	0.22	2.37	0.23	0.22
L_PI34_1	6.48	0.501	5.02	0.57	0.08
L_PI34_2	4.59	0.26	4.15	0.11	0.16
L_PI36_0	3.57	0.34	2.8	0.26	0.10
L_PI36_1	14.15	1.24	9.31	0.99	0.01
L_PI36_2	28.07	2.25	18.5	1.12	0.003
L_PI36_3	46.02	3.99	28.61	1.98	0.002
L_PI38_1	11.12	1.20	5.22	0.79	0.002
L_PI38_2	141.16	14.94	61.32	8.51	0.001
L_PI38_3	648.17	61.76	399.12	19.19	0.003
L_PI38_4	955.97	85.72	802.87	39.00	0.13
L_PI38_5	153.26	12.2	126.5	5.12	0.07
L_PI40_2	9.01	1.25	8.67	0.60	0.81
L_PI40_3	11.97	1.15	7.46	0.56	0.005
L_PI40_4	18.32	1.53	11.75	1.13	0.006
L_PI40_5	19.03	1.08	13.45	0.53	0.001
L_PI40_6	15.68	0.71	12.49	0.68	0.01
**Total PS**	**1814.09**	**445.13**	**1240.94**	**149.90**	**0.29**
L_PS32_0	14.02	1.87	12.03	1.92	0.51
L_PS32_1	nd	-	nd	-	-
L_PS34_0	10.85	1.37	9.38	0.90	0.43
L_PS34_1	25.7	5.80	17.53	2.74	0.27
L_PS34_2	5.34	0.99	4.68	0.33	0.57
L_PS36_0	25.07	5.99	15.81	1.64	0.20
L_PS36_1	135.34	33.58	83.18	10.02	0.20
L_PS36_2	36.59	8.88	24.74	2.80	0.27
L_PS36_3	15.77	3.40	13.08	1.39	0.51
L_PS38_1	nd	-	nd	-	-
L_PS38_2	30.13	7.79	16.93	1.85	0.16
L_PS38_3	162.72	42.46	107.88	14.12	0.28
L_PS38_4	570.09	135.58	449.79	52.49	0.46
L_PS38_6	72.86	15.55	63.68	6.51	0.63
L_PS40_4	44.63	11.28	29.62	4.31	0.28
L_PS40_5	140.837	37.33	81.48	11.44	0.19
L_PS40_6	524.08	135.80	311.08	47.10	0.20
L_PS42_6	nd	-	nd	-	-

## Data Availability

Data is contained within the article or Appendix A.

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
