# Peer review of "3,5-Dimethyl-2,4,6-trimethoxychalcone Lessens Obesity and MAFLD in Leptin-Deficient ob/ob Mice"

_ijms, 2024, doi:10.3390/ijms25189838_

Round 1

Reviewer 1 Report

Comments and Suggestions for Authors

Dear authors, your article is well-written, inserts in the aims and scopes of the Journal and presents some novelty. Even so, I recommend:

Mention Figure 1 in text

Table 1 and so on: insert statistic treatment

Mention each animal group in section 4.

In order to enrich the article, I think that HPLC, MALDI-TOF or RMN will be an added-value

Section 4: Animals group: … and diet? Please, give more information about each animals’ group

Lithium chloride only induces discomfort... It is not a positive controI. I strongly recommend using a positive drug already utilized to treat this MAFLD and obesity. Reformulate this-

Others:

The title needs to be formatted

(Figure 5A and B)

Insert images along with the article, not at the final of the section to be easier for the reader

GraphPad Prism 6.05 (GraphPad Software, La Jolla, CA, USA).” No?

Why do you use different colours in Table 2? Please, mention it.

Comments on the Quality of English Language

Minor editing language is required

Author Response

Dear authors, your article is well-written, inserts in the aims and scopes of the Journal and presents some novelty.

We thank the reviewer for the time spent evaluating our manuscript and his (her) positive and constructive feedback on our results. The manuscript has been corrected taking into account all the criticisms and suggestions raised with the exception of the comment concerning Figure 5A and B which remains confusing for us. Anyway, you will find our point-by-point answers.

Even so, I recommend:

Mention Figure 1 in text

The text has been amended, and Figure 1 is now mentioned in the text as follows;

“…The different steps allowing the synthesis of TriMetChalc are summarized below, and the intermediate compounds are illustrated in Figure 1A. The 1H proton and 13C carbon NMR spectra of the final product are given in Figure 1B and C respectively…”

Table 1 and so on: insert statistic treatment

Thank you for this useful comment. Statistic treatment has been added in the legend of all tables.

Mention each animal group in section 4.

As requested, more detailed information on this has been included in section 4 and the legend of each figure and table.

In section 4, the text has been modified as follows:

“…For all experiments, vehicle and TriMetChalc-treated groups each included 8 animals each. When different doses of TriMetChalc were tested, each of them has been evaluated on a group comprising 8 animals…”

In order to enrich the article, I think that HPLC, MALDI-TOF or RMN will be an added-value

We totally agree with this referee’s comment. Accordingly, we added the proton and carbon NMR plots of TriMetChalc to Figure 1_R1 (see Figure 1_R1B and C).

Section 4: Animals group: … and diet? Please, give more information about each animals’ group

As mentioned above, more detailed information regarding this has been included in the Ms.

Lithium chloride only induces discomfort... It is not a positive controI. I strongly recommend using a positive drug already utilized to treat this MAFLD and obesity. Reformulate this-

Lithium is well known to induce malaise/nausea in rodents. We used Pica behavior (Kaolin intake) in order to test the possible induction of malaise/nausea by the different doses of TriMetChalc. It seemed important to us to compare the effects of the different doses of TriMetChalc to the effect of lithium tested under the same conditions. The term positive control is clumsy, and this sentence has been changed to read:“…The effects of the different TriMetChalc doses were compared to those of lithium chloride (150 mg/kg BW), a compound well known to induce nausea and pica behavior in rodents….”

Others:

The title needs to be formatted

I'm not sure I fully understand the referee's request here. However, we have modified the title as follows:

3,5-dimethyl-2,4,6-trimethoxychalcone lessens obesity and MAFLD in leptin-deficient ob/ob mice.

(Figure 5A and B)

Insert images along with the article, not at the final of the section to be easier for the reader

Could the referee clarify his (her) request? For the insertion of all the figures, we followed the recommendations of the template and inserted the figures and tables in chronological order in the "Figures, Tables and Scheme " section.

GraphPad Prism 6.05 (GraphPad Software, La Jolla, CA, USA).” No?

That’s a right, this information has been added.

Why do you use different colours in Table 2? Please, mention it.

This point has been added in the legend of Table 2 as follows:

Fatty acids whose concentration was statistically decreased following treatment with TriMetChalc appear in green in the table, and in red for those where treatment induced a significant increase in their concentration.

Reviewer 2 Report

Comments and Suggestions for Authors

In the manuscript entitled “3,5-dimethyl-2,4,6-trimethoxychalcone lessens obesity and metabolic dysfunction-associated fatty liver disease (MAFLD) in leptin-deficient ob/ob mice” Gaige etal have identified a chalcone derivative, TriMetChalc that was found to reduce obesity and MAFLD in ob/ob mice. The data point towards TriMetChalc as an effective molecule in curbing obesity-associated MASLD. However, the study lacks what molecular targets apart from lipid metabolites are possibly regulated by TriMetChalc. Below are some concerns that if addressed, could provide a better understanding of TriMetChalc action in controlling MASLD phenotype:

1. The authors discuss that TriMetChalc may function independent of hypothalamic AMPK activation because it causes anorexigenic effect. However, since TriMetChalc reduces lipid accumulation, it would be important to know whether it regulates hepatic AMPK activation that is known to suppress lipogenesis and increase fatty acid oxidation (reference: PMC6164956). The effect of TriMetChalc on targets of AMPK such as acetyl-CoA carboxylase phosphorylation should be examined to get detailed insight into how TriMetChalc functions.

2. The data presented for PUFA does not show significant increase (p is greater than 0.05) from figure 9 and table 2, even though the authors discuss significant effects on PUFA. The PUFA to SFA ratio increases only because there is a decrease level of SFA after TriMetChalc administration. The authors need to check their data and correctly report it in the results and discussion according to what is shown in the figures.

3. The authors show that lipid droplet numbers and surface was observed in TriMetChalc mice. Since lipid droplet numbers are changed, the investigators should perform immunostaining for lipid droplet proteins such as perilipin 2 or PLIN2 to examine whether their expression is affected by TriMetChalc. These proteins are induced in lipid droplets during hepatic steatosis and hence would substantiate the results from ORO. 

4. To gain better insight into the molecular targets of TriMetChalc, the genes involved in PC synthesis should be checked to assess whether changes in PC caused by TriMetChalc are due to its effect on these genes.

Author Response

Reviewer 2

In the manuscript entitled “3,5-dimethyl-2,4,6-trimethoxychalcone lessens obesity and metabolic dysfunction-associated fatty liver disease (MAFLD) in leptin-deficient ob/ob mice” Gaige et al have identified a chalcone derivative, TriMetChalc that was found to reduce obesity and MAFLD in ob/ob mice. The data point towards TriMetChalc as an effective molecule in curbing obesity-associated MASLD. However, the study lacks what molecular targets apart from lipid metabolites are possibly regulated by TriMetChalc. Below are some concerns that if addressed, could provide a better understanding of TriMetChalc action in controlling MASLD phenotype:

We thank the reviewer for the time spent evaluating our manuscript and his (her) constructive feedback. The manuscript has been corrected, taking into account most of the criticisms and suggestions raised. We are aware that our work constitutes a first step and that the more precise identification of molecular target(s) will have to be carried in the future. That’s why, we discussed this point. However, some concerns and suggestions, although scientifically justified, seem difficult to address in the present work within the time frame allocated for the Ms revision. Below, we have answered the questions point-by-point.

  1. The authors discuss that TriMetChalc may function independent of hypothalamic AMPK activation because it causes anorexigenic effect. However, since TriMetChalc reduces lipid accumulation, it would be important to know whether it regulates hepatic AMPK activation that is known to suppress lipogenesis and increase fatty acid oxidation (reference: PMC6164956). The effect of TriMetChalc on targets of AMPK such as acetyl-CoA carboxylase phosphorylation should be examined to get detailed insight into how TriMetChalc functions.

This point raised by the referee is extremely important. Previous papers on DMC and the effects of TriMetChalc on food intake and hepatic steatosis reported here may suggest differential effects of this compound on AMPK activity at the central (hypothalamus) and peripheral (liver) levels. We will investigate this hypothesis, but we really do not wish to include these results in this present manuscript.

  1. The data presented for PUFA does not show significant increase (p is greater than 0.05) from figure 9 and table 2, even though the authors discuss significant effects on PUFA. The PUFA to SFA ratio increases only because there is a decrease level of SFA after TriMetChalc administration. The authors need to check their data and correctly report it in the results and discussion according to what is shown in the figures.

Thank you for this useful comment. As mentioned by the referee, our lipidomic analysis revealed a significant increase in linoleic and alpha-linolenic acids, but the increase of these two PUFAs species did not translate into an overall increase in PUFA contents. Accordingly, we have modified the presentation of the results to provide a more consistent description of the observed changes and weighted the discussion to clearly indicate that despite the increase in linoleic and alpha-linolenic acids the overall quantity of PUFAs generally did not increase.

See the modification in the results section:

“…Regarding PUFAs, linoleic acid (FA C18:2n6) and alpha-linolenic acid (FA C18:3n3) were found to be increased after TriMetChalc treatment (Table 2). Accordingly, total conjugated linoleic acid (CLA-FA) was increased in the TriMetChalc group, although total PUFAs content was not statistically increased in response to TriMetChalc treatment (Figure 9I-J and Table 2)…”

And in the discussion:

“…Our lipidomic analyses revealed a significant increase in linoleic acid (FA C18:2w6) and alpha-linolenic acid (FA C18:3w3). However, this increase limited to these two PUFAs species did not translate into an overall increase in PUFAs contents. However, considering the decreased level of SFA after TriMetChalc administration, the PUFAs/SFAs ratio was found to be increased by TriMetChalc administration… 

…. Although the overall PUFA concentration was not modified by TriMetChalc treatment, the increase in linoleic and alpha-linolenic acids PUFAs and the concomitant reduction in SFA and MUFA induced by TriMetChalc should be highlighted and could contribute to improving the liver status of the animals…”

Round 2

Reviewer 1 Report

Comments and Suggestions for Authors

Thanks for the alterations 

Reviewer 2 Report

Comments and Suggestions for Authors

None